# Ensembled deep learning model outperforms human experts in diagnosing biliary atresia from sonographic gallbladder images

Wenying Zhou [1,12], Yang Yang [2,12], Cheng Yu [3,12], Juxian Liu [4,12], Xingxing Duan [5], Zongjie Weng[6], Dan Chen [7], Qianhong Liang[8], Qin Fang [9], Jiaojiao Zhou [4], Hao Ju [10], Zhenhua Luo [11], Weihao Guo[1], Xiaoyan Ma[7], Xiaoyan Xie [1,13✉], Ruixuan Wang [2,13✉] & Luyao Zhou [1,13✉]

It is still challenging to make accurate diagnosis of biliary atresia (BA) with sonographic gallbladder images particularly in rural area without relevant expertise. To help diagnose BA based on sonographic gallbladder images, an ensembled deep learning model is developed. The model yields a patient-level sensitivity 93.1% and specificity 93.9% [with areas under the receiver operating characteristic curve of 0.956 (95% confidence interval: 0.928-0.977)] on the multi-center external validation dataset, superior to that of human experts. With the help of the model, the performances of human experts with various levels are improved. Moreover, the diagnosis based on smartphone photos of sonographic gallbladder images through a smartphone app and based on video sequences by the model still yields expert-level performances. The ensembled deep learning model in this study provides a solution to help radiologists improve the diagnosis of BA in various clinical application scenarios, particularly in rural and undeveloped regions with limited expertise.

[1] Department of Medical Ultrasonics, Institute for Diagnostic and Interventional Ultrasound, The First Affiliated Hospital, Sun Yat-sen University, Guangzhou, P. R. China. [2] School of Computer Science and Engineering, Sun Yat-sen University, Guangzhou, P. R. China. [3] Department of Ultrasound, Union Hospital, Tongji Medical College, Huazhong University of Science and Technology, Wuhan, P. R. China. [4] Department of Ultrasound, West China Hospital, Sichuan University, Chengdu, P. R. China. [5] Department of Ultrasound, Hunan Children's Hospital, Changsha, P. R. China. [6] Department of Medical Ultrasonics, Fujian Provincial Maternity and Children's Hospital, Affiliated Hospital of Fujian Medical University, Fuzhou City, P. R. China. [7] Department of Ultrasound, Guangdong Women and Children' Hospital, Guangzhou, P. R. China. [8] Department of Ultrasound, Hexian Memorial Affiliated Hospital of Southern Medical University, Guangzhou, P. R. China. [9] Department of Ultrasound, The First People's Hospital of Foshan, Foshan City, P. R. China. [10] Department of Ultrasound, Shengjing Hospital of China Medical University, Shenyang, P. R. China. [11] Institute of Precision Medicine, The First Affiliated Hospital, Sun Yat-sen University, Guangzhou, P. R. China. [12] These authors contributed equally: Wenying Zhou, Yang Yang, Cheng Yu, Juxian Liu. [13] These authors jointly supervised this work: Xiaoyan Xie, Ruixuan Wang, Luyao Zhou. ✉email: xiexyan@mail.sysu.edu.cn; wangruix5@mail.sysu.edu.cn; zhouly6@mail.sysu.edu.cn

Biliary atresia (BA) is a rare disease of infancy that affects both intrahepatic and extrahepatic bile ducts[1], with the prevalence rate of about 1 in 5000–19,000 infants all over the world[2–6]. It is the most common cause for liver transplantation in infants aged <1 year[7]. Optimal clinical outcome often needs timely diagnosis and Kasai portoenterostomy (KPE) surgery before age 2 months, which is associated with longer native liver survival[8–10]. However, early identifying BA remains challenging in infants with cholestasis. Researchers have endeavored to screen the direct bilirubin concentration[11,12] or stool color[4,10] in newborns and infants for early identification of BA and showed promising results (with sensitivities of 97.1–100%). Recently, serum matrix metalloproteinase-7 was reported as an effective diagnostic biomarker for BA, with sensitivity of 97.0–98.7%[13,14]. However, these tests are high resource-consumed and might be impractical in many countries and areas with underdeveloped healthcare conditions.

Ultrasound (US) examination, due to its radiation-free and low-cost noninvasive property, is still the most widely used method for initial detection of BA in jaundiced infants particularly in developing Asian countries like China and India[15–18]. Gallbladder abnormality is one of the most popular sonographic features used to identify BA[19–22]. As previously reported, gallbladder abnormalities can yield both sensitivities and specificities >90% in an experienced hand in the diagnosis of BA[23]. However, it is still difficult to make a correct diagnosis by US examination mainly due to the lack of expertise in both diagnosis and management of BA in most hospitals particularly located in underdeveloped regions. Consequently, a substantial proportion of potential BA patients are often misdiagnosed followed by inappropriate treatments, and the average age of BA patients at KPE surgery was delayed being >70 days in China[24].

To improve the accuracy of US diagnosis of BA in underdeveloped countries or regions, one potentially promising way is to make use of the artificial intelligence (AI) techniques. Among the AI techniques, deep learning models, particularly the convolutional neural networks (CNNs), have been shown superior or comparable to human experts in many medical data analysis tasks, such as the diagnosis of skin cancers, localization and identification of polyps, axillary lymph node status in early-stage breast cancer, and lung cancer screening[25–31]. However, as far as we know, no AI model based on sonographic images has been developed for the diagnosis of BA. Considering the fact that US examination is very common in both primary and tertiary hospitals in China, any well-developed AI model based on sonographic gallbladder images would alleviate the shortage of expertise in primary hospitals and may improve the diagnostic accuracy of the rare disease.

The purpose of this study was to develop an ensembled deep learning model (EDLM) for automatically and accurately identifying BA in infants with conjugated hyperbilirubinemia, based on limited number of sonographic gallbladder images collected from multiple centers and to help doctors improve their diagnosis of BA. The EDLM yields a patient-level sensitivity 93.3% and specificity 85.2% on the internal validation dataset, and sensitivity 93.1% and specificity 93.9% [with area under the receiver operating characteristic curve (AUC) of 0.956 (95% confidence interval: 0.928–0.977)] on the multi-center external validation dataset, superior to that of human experts. With the help of the model, the performances of human experts with various levels are improved. Moreover, the diagnosis based on smartphone photos of sonographic gallbladder images through a smartphone app and based on video sequences by the model still yields expert-level performance. The EDLM provides a solution to help radiologists improve the diagnosis of BA in various clinical application scenarios, particularly in rural and undeveloped regions with limited expertise.

## Results

**Internal evaluation of the ensemble deep learning approach.** The ensemble deep learning approach was first evaluated in a fivefold cross-validation manner on the training cohort. Specifically, the training cohort was partitioned into five complementary subsets of an equivalent number of patients. Then, every time four of the subsets were used as a training dataset to train an ensembled deep learning model, and the ensembled model was then applied to predict the category of each image in the remaining one (testing) subset. Such a process was repeated five times, each time using a unique subset as the testing dataset.

At both the image level and the patient level, the EDLM outperformed the two experts in diagnosing BA, with the image-level sensitivity 88.2%, specificity 89.8%, and accuracy 89.4% of the model versus the sensitivity 93.8%, specificity 53.7%, and accuracy 63.7% of the most experienced expert, and the patient-level sensitivity 93.3% and specificity 85.2% of the model versus the sensitivity 90.0% and specificity 57.6% of the most experienced expert (Table 1). The receiver operating characteristic (ROC) curves of the model at both levels also confirmed its superior performance over human experts [AUC of 0.952 versus 0.738 and 0.837 at the image level, 0.953 versus 0.738 and 0.813 at the patient level, respectively] (Fig. 1). The $\kappa$ value of the agreement between the two human experts in the identification of BA was 0.358 at the image level and 0.306 at the patient level.

**Robustness of the AI models to various scanning conditions.** Considering that the trained deep learning model could be

**Table 1 The diagnostic performance of the ensembled deep learning model (in a cross-validation manner) and two human experts on the internal dataset.**

|  | AUC | Sensitivity (%) | Specificity (%) | Accuracy (%) | PPV (%) | NPV (%) | P value[a] |
|---|---|---|---|---|---|---|---|
| **Image level** | | | | | | | |
| AI Model | 0.952 (0.945, 0.959) | 88.2 (86.0, 90.2) | 89.8 (88.6, 90.9) | 89.4 | 74.1 | 95.8 | — |
| Expert A | 0.837 (0.825, 0.849) | 76.3 (73.4, 79.0) | 91.0 (89.9, 92.1) | 87.4 | 73.9 | 92.0 | <0.001 |
| Expert B | 0.738 (0.723, 0.752) | 93.8 (92.1, 95.3) | 53.7 (51.8, 55.5) | 63.7 | 40.3 | 96.3 | <0.001 |
| **Patient level** | | | | | | | |
| AI Model | 0.953 (0.939, 0.964) | 93.3 (90.1, 95.8) | 85.2 (82.6, 87.6) | 87.6 | 72.0 | 96.9 | — |
| Expert A | 0.813 (0.789, 0.835) | 65.8 (60.4, 70.9) | 96.8 (95.3, 97.9) | 87.8 | 89.3 | 87.4 | <0.001 |
| Expert B | 0.738 (0.711, 0.763) | 90.0 (86.2, 93.0) | 57.6 (54.1, 61.0) | 67.0 | 46.8 | 93.4 | <0.001 |

Note: 95% confidence intervals are included in brackets. Source data are provided as a Source data file.
*AI* artificial intelligence, *AUC* area under receiver operating characteristic curve, *PPV* positive predictive value, *NPV* negative predictive value.
[a]The P values were from the comparison between the AUC of the ensemble deep learning model and the AUCs of two human experts. Differences between various AUCs were compared using a Delong test.

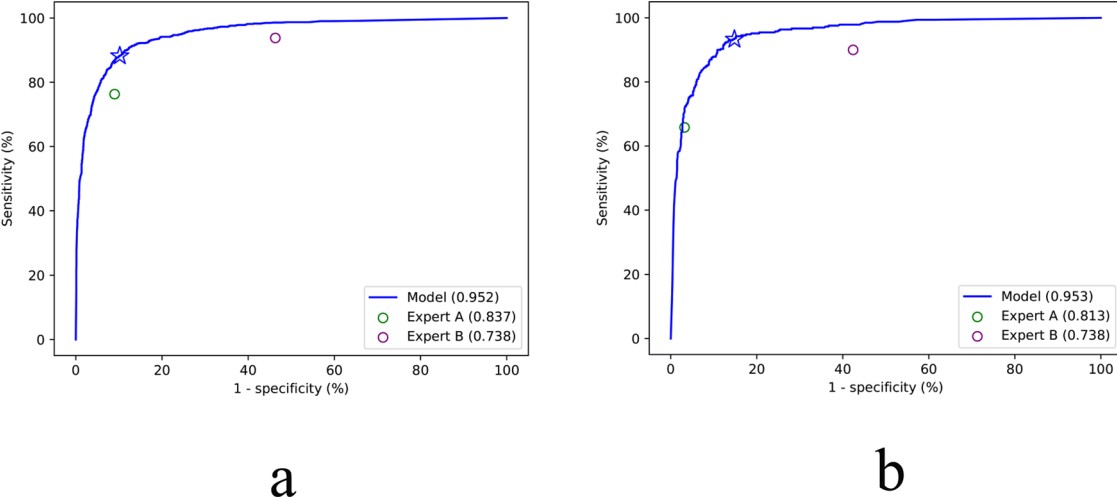

**Fig. 1 The ROC curves of the ensembled deep learning models for the diagnosis of biliary atresia on the internal cross-validation dataset with two human experts' performance for comparison. a** The ROC curve of the model at the image level. **b** The ROC curve of the model at the patient level. The performance of the two experts is represented by individual solid circle, which is inferior to that of the ensembled deep learning model. The blue star represents the performance of the model with the default threshold (0.5) to binarize outputs of the model. Source data are provided as a Source data file. ROC receiver operating characteristic.

deployed to various hospitals in which US scanning conditions might be different from those of the data for model training, the ensemble deep learning approach was also evaluated in terms of its robustness to screening machines, transducer frequencies, and scanning period. The same training procedure was applied to train each ensemble model as for the internal evaluation.

In the training cohort, images were retrospectively divided into three subsets based on whether the images were obtained from machines of brand Mindray, Supersonic, or the others (including TOSHIBA, Siemens, Samsung, HITACHI, ALOKA, Philips, GE, and Esaote) or into two subsets based on whether the images were obtained by transducers of frequencies ≥14 MHz or by transducers of frequencies <14 MHz or into two subsets based on whether the images were obtained before year 2018 or thereafter. For each scanning factor, with every unique subset of images as the validation dataset and the remaining subset(s) as the training dataset, the sensitivity of the trained EDLM was roughly in between those of two human experts (Supplementary Tables 1 and 2 and Supplementary Figs. 1 and 2), supporting that the EDLMs were robust enough to be deployable to different medical centers and for different screening machines. Furthermore, when using images in relatively moderate quality (with frequency <14 MHz, scanning period ≤2018, Supersonic + others or Mindray + others) to train the model and using the remaining subset (s) for validation, we found that the diagnostic performance of the model was higher than that when the training set and the testing set were reversed (AUC 0.931 versus 0.835 for transducer frequency, 0.900 versus 0.832 for screening time, 0.950 or 0.807 versus 0.787 for screening machine, respectively at the image level; Supplementary Table 1), which indicates that the EDLM might still work well for new screening machines, which often generate images in higher quality.

**External validation of the EDLM.** More strictly, the effectiveness of the EDLM was evaluated by external validation with US images obtained from the other six hospitals. The EDLM yielded an image-level accuracy 92.3%, sensitivity 88.6%, specificity 93.7%, positive predictive value 84.6%, and negative predictive value 95.5%, respectively, clearly outperforming the three experts whose diagnosis sensitivities were 77.1, 69.5, and 87.3%, and specificities are 83.5, 90.2, and 90.2%, respectively (Table 2, rows 1–4). The

superior performance of the model could also be seen from the ROC curve of the EDLM (AUC 0.942 versus 0.803, 0.799 and 0.888, all $P < 0.05$; Fig. 2a). Specifically, there were 20 images misdiagnosed by three experts but correctly diagnosed by the model. Of the 20 images, 12 images were false negatives (BA misdiagnosed as non-BA) and 8 images were false positives (non-BA misdiagnosed as BA). On the contrary, there were also 20 images misdiagnosed by the model but correctly assessed by three experts on the external validation dataset, of which 14 images were false positives and 6 images were false negatives. Overall, experts missed more BA cases than the AI model.

When using the majority vote over the predicted classes of multiple images for each patient, the EDLM achieved an accuracy 93.6%, sensitivity 93.1%, specificity 93.9%, positive predictive value 88.8%, and negative predictive value 96.3% (Table 2, row 8). Another way to obtain patient-level performance was from the diagnosis of a single image for each patient, where the single image was chosen from the multiple images of the patient by a radiologist based on the imaging quality (e.g., choosing the images with clear contour of gallbladder, less blurry, better view of gallbladder, etc.). Such single-image diagnosis by the EDLM achieved an accuracy 91.6%, sensitivity 87.3%, specificity 93.9%, positive predictive value 88.1%, and negative predictive value 93.4% (Table 2, row 15). Both performances of the majority vote and the single-image based diagnosis at the patient level by the EDLM outperformed those of all the three experts (all $P < 0.05$), as seen in Table 2 (rows 9–11 and rows 16–18) and in the ROC curve (Fig. 2b, c). In addition, the $\kappa$ value of the agreement between the three experts in the identification of BA ranged from 0.570 to 0.656 at the image level, 0.592 to 0.683 at the patient level diagnosed with all images, and 0.603 to 0.672 at the patient level diagnosed with single image.

**Combination of the diagnosis from the EDLM and expert.** Considering the potentially serious consequence of delayed treatment for infants with BA, it is desirable to improve the sensitivity of diagnosis while keeping the specificity at a high level. One possible way to achieve such a goal is to combine the diagnosis of a human expert with that of the deep learning model. Here on the external validation dataset, each patient was diagnosed with BA if either an expert or the EDLM thought so. With

**Table 2 The diagnostic performance of the ensembled deep learning model, three experts, and the human–AI combination on the external validation dataset and the comparisons between the model and three experts based on sonographic videos.**

| | | AUC | Sensitivity (%) | Specificity (%) | Accuracy (%) | PPV (%) | NPV (%) | P value[a] |
|---|---|---|---|---|---|---|---|---|
| Image level | AI Model | 0.942 (0.924, 0.957) | 88.6 (83.8, 92.3) | 93.7 (91.5, 95.5) | 92.3 | 84.6 | 95.5 | — |
| | Expert C | 0.803 (0.774, 0.829) | 77.1 (71.2, 82.3) | 83.5 (80.3, 86.3) | 91.7 | 64.5 | 90.3 | <0.001 |
| | Expert D | 0.799 (0.770, 0.825) | 69.5 (63.2, 75.3) | 90.2 (87.6, 92.5) | 84.4 | 73.5 | 88.3 | <0.001 |
| | Expert E | 0.888 (0.864, 0.908) | 87.3 (82.4, 91.3) | 90.2 (87.6, 92.5) | 89.4 | 77.7 | 94.8 | <0.001 |
| | Expert C + Model | — | 95.3 (91.8, 97.7) | 80.5 (77.1, 83.6) | 84.7 | 65.6 | 97.8 | — |
| | Expert D + Model | — | 93.2 (89.2, 96.1) | 86.1 (83.1, 88.8) | 88.1 | 72.4 | 97.0 | — |
| | Expert E + Model | — | 95.8 (92.3, 97.9) | 86.1 (83.1, 88.8) | 88.8 | 72.9 | 98.1 | — |
| Patient level (diagnosed with all images) | AI Model | 0.956 (0.928, 0.977) | 93.1 (86.4, 97.2) | 93.9 (89.5, 96.8) | 93.6 | 88.8 | 96.3 | — |
| | Expert C | 0.831 (0.784, 0.872) | 88.2 (80.4, 93.8) | 78.1 (71.6, 83.6) | 81.5 | 67.7 | 92.7 | <0.001 |
| | Expert D | 0.843 (0.797, 0.882) | 81.4 (72.4, 88.4) | 87.2 (81.7, 91.6) | 85.2 | 76.9 | 90.0 | <0.001 |
| | Expert E | 0.895 (0.855, 0.927) | 89.2 (81.5, 94.5) | 89.8 (84.7, 93.7) | 89.6 | 82.0 | 94.1 | 0.013 |
| | Expert C + Model | — | 96.1 (90.3, 98.9) | 75.5 (68.9, 81.4) | 82.6 | 67.1 | 97.4 | — |
| | Expert D + Model | — | 97.1 (91.6, 99.4) | 83.2 (77.2, 88.1) | 87.9 | 75.0 | 98.2 | — |
| | Expert E + Model | — | 98.0 (93.1, 99.8) | 85.2 (79.4, 89.9) | 89.6 | 77.5 | 98.8 | — |
| Patient level (diagnosed with single image) | AI Model | 0.930 (0.894, 0.956) | 87.3 (79.2, 93.0) | 93.9 (89.5, 96.8) | 91.6 | 88.1 | 93.4 | — |
| | Expert C | 0.796 (0.745, 0.840) | 76.5 (67.0, 84.3) | 82.7 (76.6, 87.7) | 80.5 | 69.6 | 87.1 | <0.001 |
| | Expert D | 0.815 (0.766, 0.857) | 70.6 (60.7, 79.2) | 92.3 (87.7, 95.7) | 85.2 | 82.8 | 85.8 | <0.001 |
| | Expert E | 0.863 (0.819, 0.900) | 83.3 (74.7, 90.0) | 89.3 (84.1, 93.2) | 86.6 | 80.2 | 92.1 | 0.016 |
| | Expert C + Model | — | 95.1 (88.9, 98.4) | 81.6 (75.5, 86.8) | 86.2 | 72.9 | 97.0 | — |
| | Expert D + Model | — | 92.2 (85.1, 96.6) | 88.3 (82.9, 92.4) | 89.6 | 80.3 | 95.6 | — |
| | Expert E + Model | — | 96.1 (90.3, 98.9) | 86.2 (80.6, 90.7) | 89.6 | 78.4 | 97.7 | — |
| Smartphone image | AI Model | 0.902 (0.863, 0.933) | 89.2 (81.5, 94.5) | 85.7 (80.0, 90.3) | 86.9 | 76.5 | 93.9 | 0.159 |
| Video[b] | AI Model | 0.941 (0.803, 0.993) | 94.1 (71.3, 99.9) | 94.1 (71.3, 99.9) | 94.1 | 94.1 | 94.1 | — |
| | Expert C | 0.794 (0.621, 0.913) | 100 (80.5, 100.0) | 58.8 (32.9, 81.6) | 79.4 | 70.8 | 100 | 0.105 |
| | Expert D | 0.912 (0.763, 0.981) | 82.4 (56.6, 96.2) | 100 (80.5, 100.0) | 91.2 | 100 | 85.0 | 0.686 |
| | Expert E | 0.941 (0.803, 0.993) | 94.1 (71.3, 99.9) | 94.1 (71.3, 99.9) | 94.1 | 94.1 | 94.1 | >0.999 |

Performance of the model on smartphone images was also included. Note: 95% confidence intervals are included in brackets. Source data are provided as a Source data file.
AI artificial intelligence, AUC area under receiver operating characteristic curve, PPV positive predictive value, NPV negative predictive value.
[a]The P values were from the comparison between the AUC of the ensemble deep learning model and the AUCs of three human experts on the same dataset. Differences between various AUCs were compared using a Delong test.
[b]Thirty-four sonographic videos were obtained from 34 infants (17 with BA and I7 without BA).

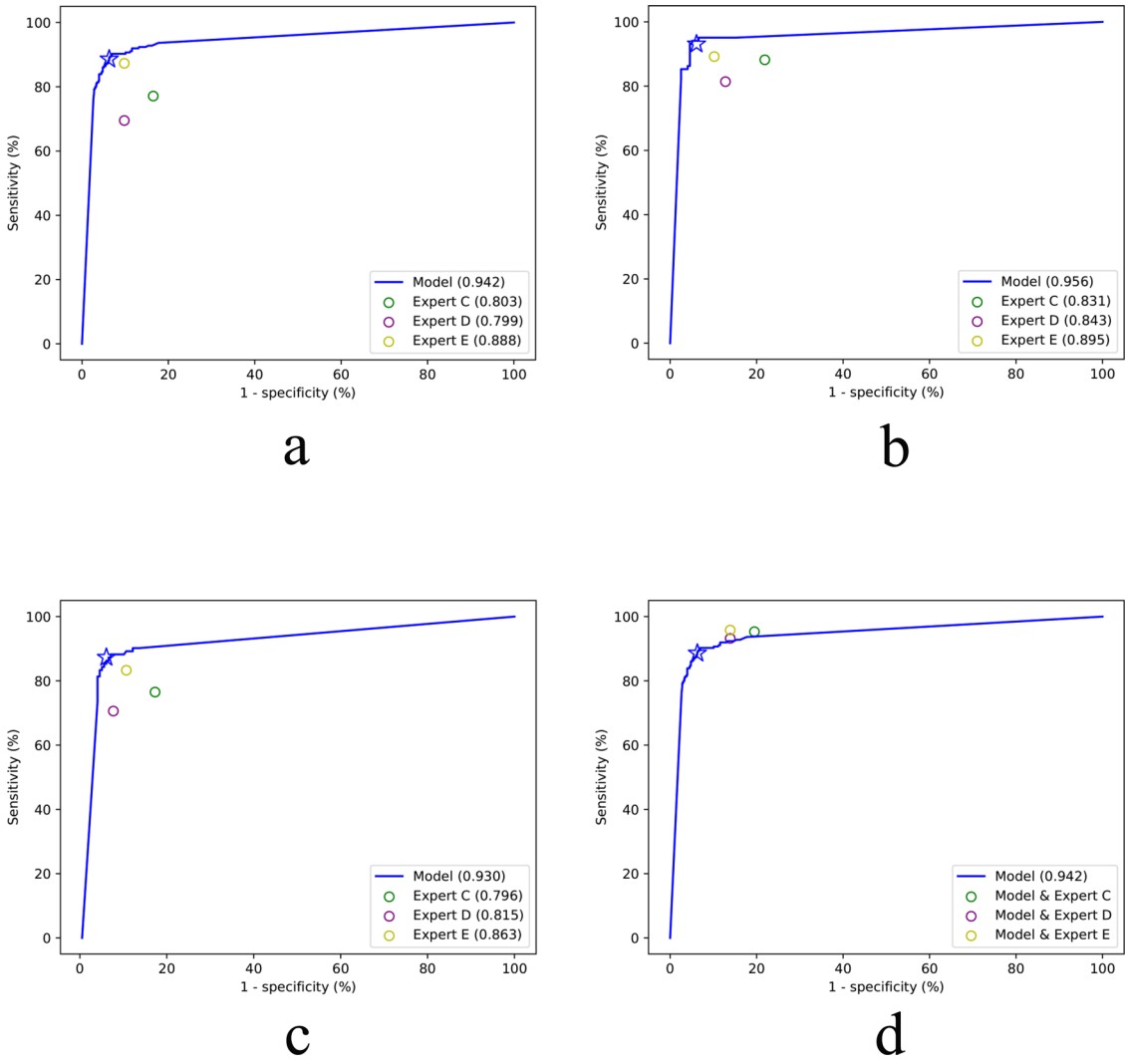

**Fig. 2 The performance of the ensembled deep learning model, human experts, and the combinations of model and humans for the diagnosis of biliary atresia on the external validation dataset. a** The ROC curve of the model at the image level. **b** The ROC curve of the model at the patient level based on majority vote. **c** The ROC curve of the model at the patient level based on single image with best image quality for each patient. **d** The performance of the combined deep learning model and human expert (circles) at the image level. The blue star represents the performance of the model with the default threshold (0.5) to binarize outputs of the model. Source data are provided as a Source data file. ROC receiver operating characteristic.

such combined diagnosis, the sensitivities of three human experts at the image level were improved substantially (Expert C, increased from 77.1% to 95.3%; Expert D, increased from 69.5% to 93.2%; Expert E, increased from 87.3% to 95.8%), although the specificities of their diagnosis decreased moderately (Expert C, decreased from 83.5% to 80.5%; Expert D, decreased from 90.2% to 86.1%; Expert E, decreased from 90.2% to 86.1%) (Table 2, rows 5–7). Similar findings were obtained when tested at the patient level with multiple images (Table 2, rows 12–14) and at the patient level with a single image (Table 2, rows 19–21). These findings suggest that the combined approach outperforms not only each expert but also the EDLM in sensitivity, as confirmed from the ROC curve in Fig. 2d (also see Supplementary Fig. 3), particularly in reducing the misdiagnosis of BA.

**Diagnosis based on smartphone photos of sonographic images by the EDLM.** In reality, the sonographic machines used for medical examination in hospitals are often not connected to the internet, and it may not be convenient or allowed to extract the original US images from the machine system. To avoid such

obstacle when applying the deep learning model in many medical centers particularly from rural areas, one simple solution is to take a photograph of the sonographic image by a smartphone and then send the photo to a remotely located AI system for intelligent diagnosis. However, the image quality of the photograph would be inevitably affected by this imaging process, e.g., with more noise included or shape and texture of gallbladder regions deformed (Fig. 3a right, Fig. 3b right). It would be desirable if the deep learning model could still work well when applied to the analysis of such smartphone photos.

To evaluate the robustness of the EDLM in this case, one original image per patient (as mentioned above for the single-image diagnosis) from the external validation dataset was pictured by a smartphone (HUAWEI P10, Rear Camera: 12 million pixels; Fig. 3a, b), with the original image information kept as much as possible during picturing (e.g., by making camera viewing direction perpendicular to the machine screen). Smartphone photos were saved in the JPEG format. As done for the original images, the region of gallbladder was extracted from each photograph and then fed into the EDLM for intelligent diagnosis.

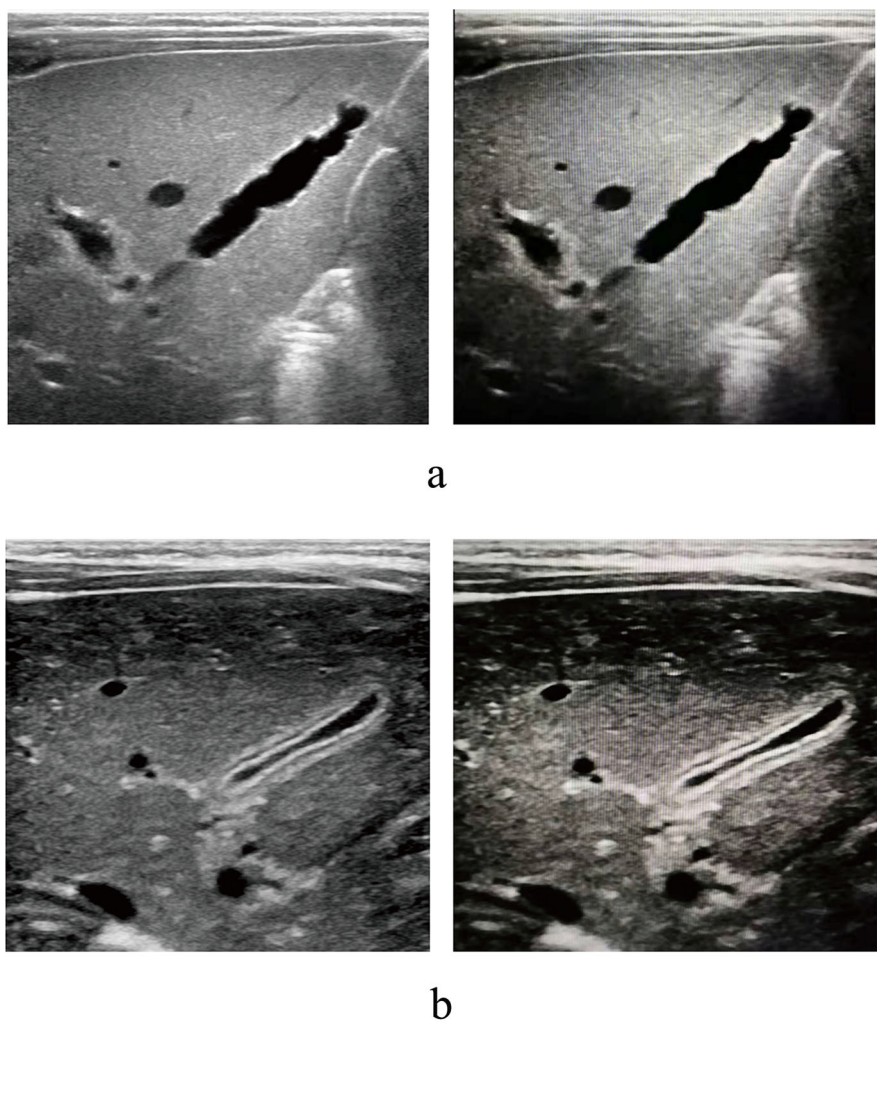

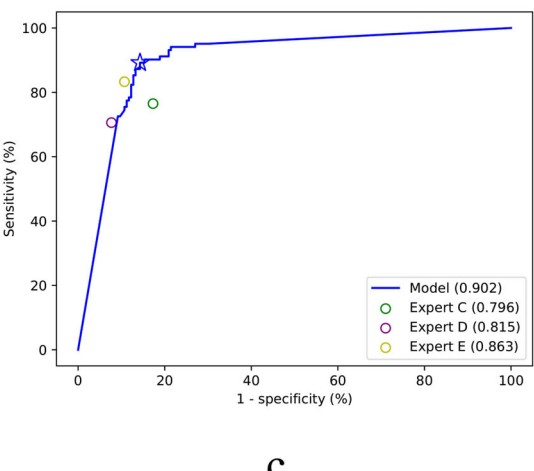

**Fig. 3 Diagnosis based on smartphone photos of sonographic images by the deep learning model. a** An exemplar original image from a patient with biliary atresia (left) and the smartphone photo of the image (right). **b** An exemplar original image from a patient without biliary atresia (left) and the smartphone photo of the image (right). **c** The receiver operating characteristic curve of the model for the diagnosis of biliary atresia on the smartphone images of external validation dataset, with three human experts' performances on the original clean external validation dataset for comparison. Source data are provided as a Source data file.

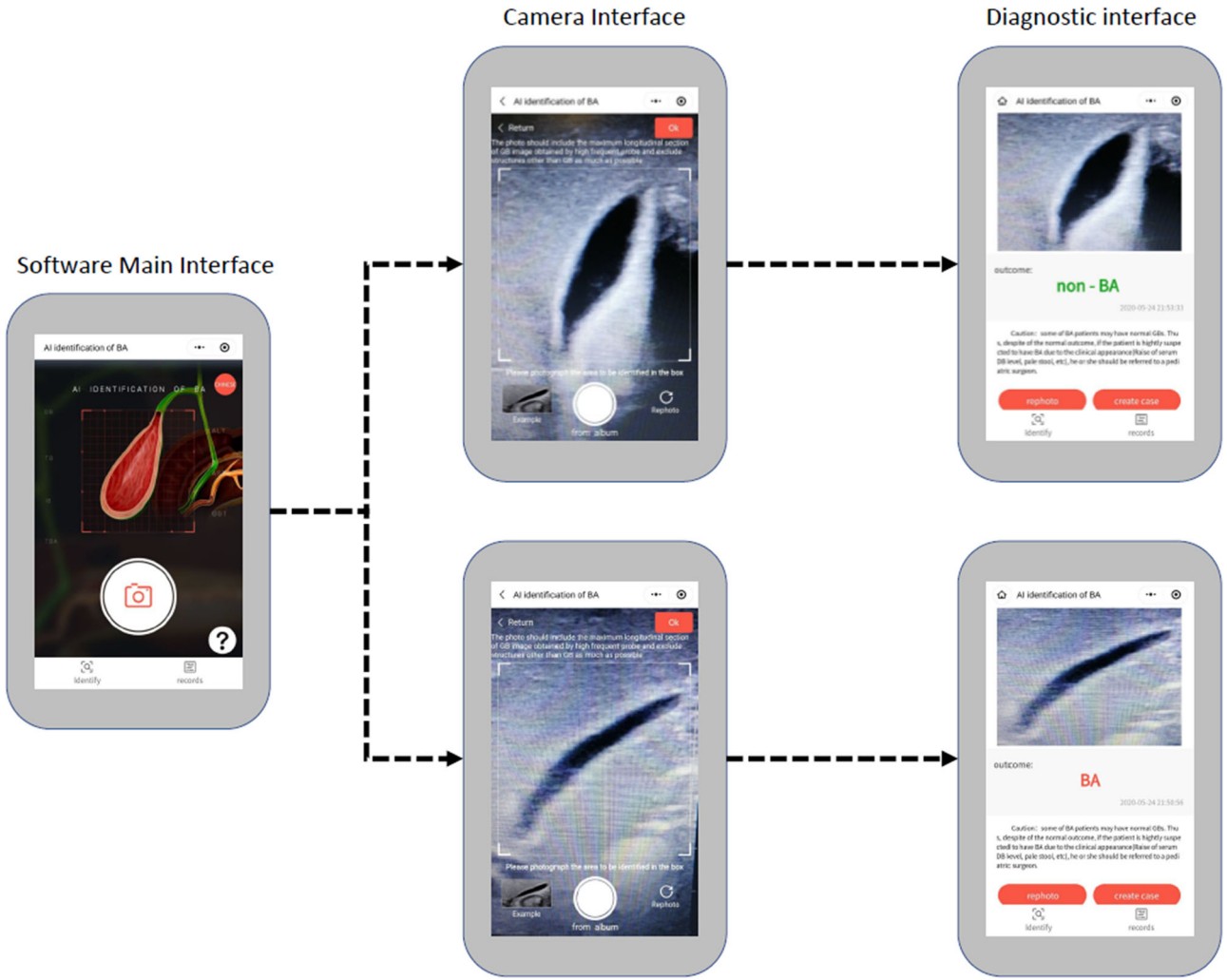

**Fig. 4 The user interface for each step of the smartphone app.** Firstly, the users open the app and take a photo of gallbladder from the screen of a sonographic machine. The photo should include the maximum longitudinal section of gallbladder image obtained by high frequent probe and exclude structures other than gallbladder as much as possible. The users then press "Ok" to send the photo to a cloud platform. The outcome of BA or non-BA will show on the smartphone screen in a few seconds. BA biliary atresia.

Although the EDLM was trained with the original clean images from the training cohort, the performance of the smartphone images by the model resulted in an accuracy 86.9%, sensitivity 89.2%, and specificity 85.7% (Table 2, row 22). The AUC value (0.902) was slightly lower than that tested with the original images (AUC = 0.930), with no statistical difference between the two ($P = 0.159$), but the ROC curve (Fig. 3c) together with the prediction performance (Table 2, row 22) suggested that such performance was still comparable to the best performance of the human expert (AUC: 0.902 versus 0.863, $P = 0.163$) and outperformed those of the other two human experts (AUC: 0.902 versus 0.796 and 0.815, both $P < 0.05$) who made diagnoses based on the original clean images.

Considering the promising external validation result based on smartphone photos, a smartphone app was developed and released (Fig. 4), from which users could freely upload photos of US images and interactively locate the gallbladder regions. The software would send photos to and collect prediction results from a cloud platform running the EDLM. An initial prospective study (sonographic gallbladder images were from multicenter photographed by different radiologists) with 71 BA patients and 103 non-BA patients (1 photo per patient) showed that the app performed similarly well, with an accuracy 85.6%, sensitivity

85.9%, specificity 85.4%, and AUC value 0.856. The small variation in performances between the prospective study and the above external validation was probably due to the uncontrolled picturing conditions in the prospective study, where different users might use different smartphones in varying lighting environments. Such smartphone app provides the opportunity to help clinicians improve their diagnostic performance particularly for hospitals in rural areas.

**Diagnosis based on sonographic videos by the EDLM.** In practice, human radiologists make diagnoses not based on observing one or a few static sonographic images but by dynamically observing the gallbladder region with real-time US scanning. Also, it would be inconvenient for radiologists to select one or a few static images and then draw bounding boxes surrounding the gallbladder before sending the images to the intelligent diagnosis system. Therefore, it would be ideal if the intelligent diagnosis system can make fully automatic diagnosis just based on the recorded video sequences of sonographic images. To achieve this goal, we trained an auto segmentation model and an initial prospective study was performed with a collection of 34 sonographic videos obtained from 34 infants (17 with BA and 17 without BA). The diagnostic performance of EDLM was

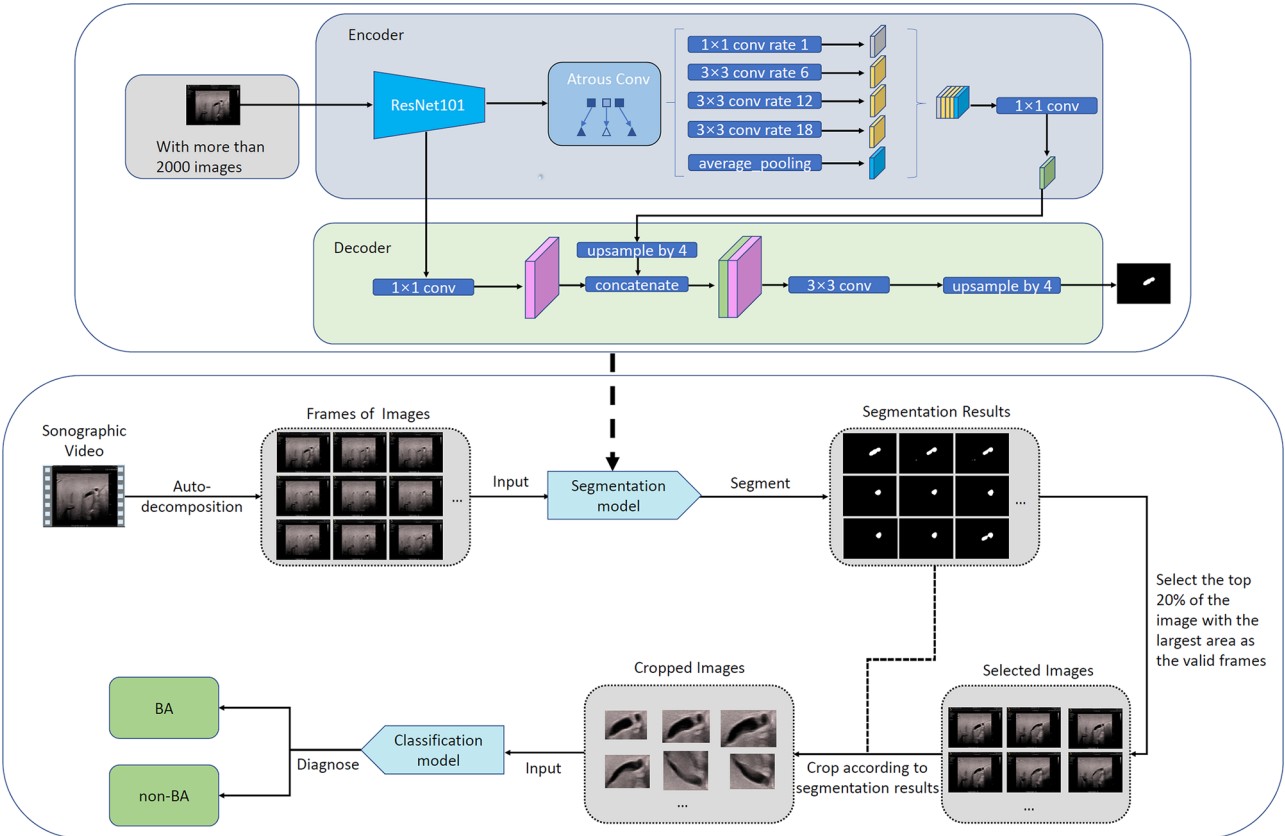

**Fig. 5 The diagnostic process for each sonographic gallbladder video by the ensembled deep learning model.** BA biliary atresia.

compared with performances of three human experts, each of whom independently made diagnoses by reviewing videos and was blinded to other clinical information.

Since only part of the images in each video contains clear gallbladder regions and the intelligent diagnosis system make decisions based on the gallbladder image region only, such images and corresponding gallbladder regions need to be automatically selected and localized for automatic diagnosis of BA from each video. The automatic localization of gallbladder regions was obtained by the well-known semantic image segmentation model DeepLab[32]. The segmentation model was trained on 2383 sonographic images, which were randomly selected from the original 3705 training images and then annotated by roughly drawing the boundary of the gallbladder regions. For each video, the trained segmentation model was applied on each video frame, and 20% frames with relatively large segmented gallbladder regions were selected for intelligent diagnosis. The rectangular region tightly containing the segmented largest gallbladder region was cropped from each selected frame and then sent to the intelligent diagnosis system for BA diagnosis. The video was diagnosed as BA if >20% selected frames were diagnosed as BA by the EDLM (Fig. 5).

Based on the fully automatic diagnosis process, 16 out of the 17 BA videos were correctly diagnosed as BA (sensitivity 94.1%), and 16 out of the 17 non-BA videos were correctly diagnosed as non-BA (specificity 94.1%). Compared to the diagnostic performances from the experts (Table 2, last 3 rows), the EDLM was comparable to three experts (all $P > 0.05$). More evaluations showed that the diagnostic performance of the EDLM changed little when the model hyperparameters varied, such as changing the percentage of the selected images from 20 to 10% and changing the percentage of diagnosed BA images from 10 to 30%, suggesting strong robustness of the EDLM.

**Initial attempt to interpret AI diagnosis**. One widely used method to interpret the black-box AI diagnosis is the class activation map (CAM), which can provide the attended image region(s) for each specific prediction from the model[33]. Based on the attended region from CAM, people may infer why the model makes the current prediction for each image (e.g., "because the model focuses on the gallbladder region and therefore uses the visual features within this region to make the decision"). If the attended region obtained by CAM covers or partly covers the regions used by human experts for diagnosis ("Consistent" in Supplementary Table 3), it may improve the sense of trust in the AI model for the current diagnosis. Otherwise, if the attended region obtained by CAM does not cover any region of interest used by experts ("Inconsistent" in Supplementary Table 3), this may indicate that the AI model does not use appropriate visual features to make current (either correct or incorrect) decision. For each image in the external validation dataset, there were five activation maps generated by five individual models within the EDLM. Within the activation maps whose associated individual models had the same classification result as that of the EDLM, the activation map that had the highest mean activation was selected for consistency assessment in comparison with human experts. Of all the external validation images, detailed inspection showed that 99.0% were consistent in decision-making between the model and human experts. Of the correctly diagnosed external validation images, 99.5% were consistent (Supplementary Table 3, row 3; also see Fig. 6a) and 0.5% were inconsistent (Supplementary Table 3, row 3; Fig. 6b); Of the incorrectly diagnosed external validation images, 100% were consistent (Supplementary Table 3, row 6; Fig. 6c).

## Discussion
In this multicenter study, we trained and validated a state-of-the-art EDLM for the diagnosis of BA based on sonographic

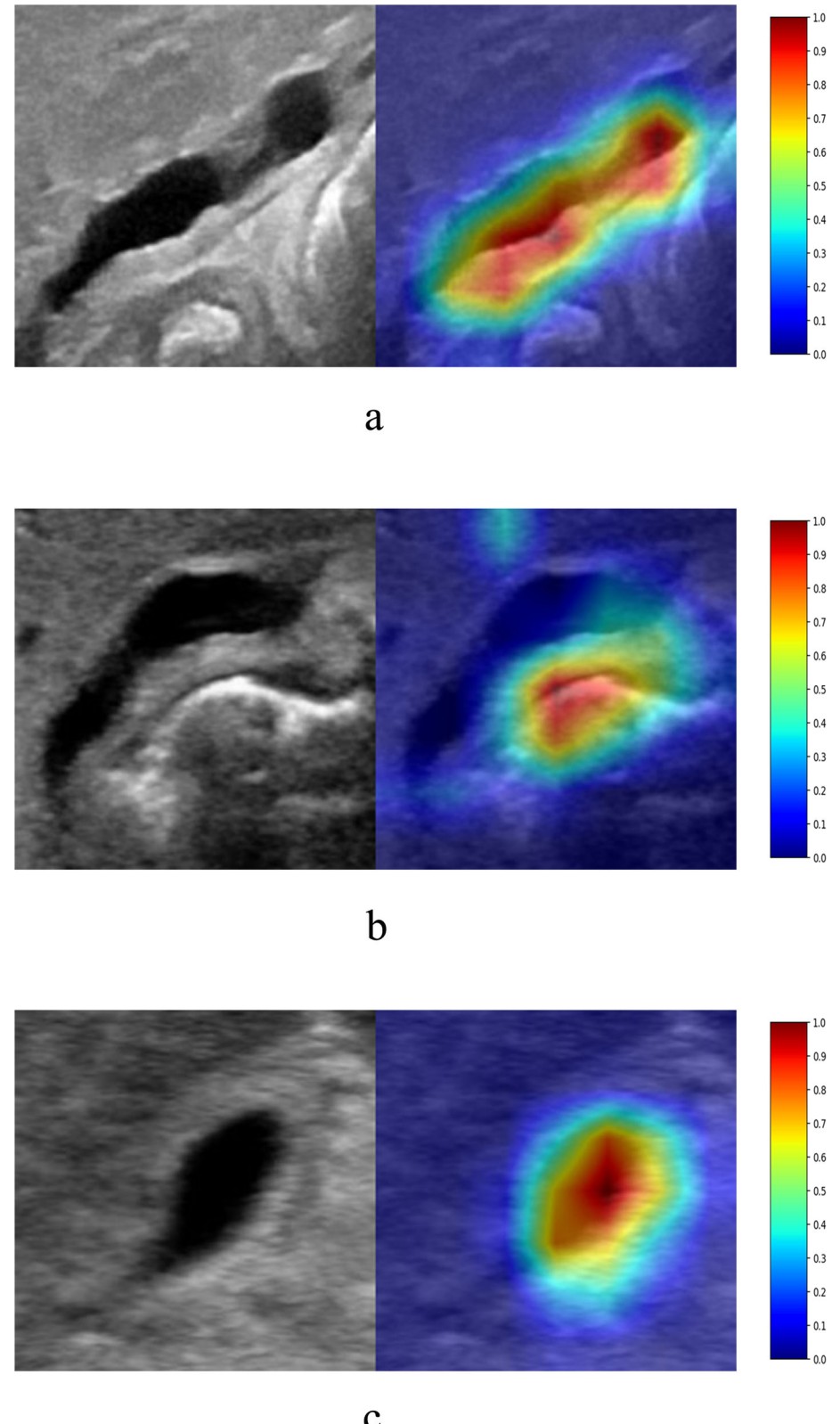

**Fig. 6 The attended regions obtained by the class activation map during diagnosis by the individual model that was within the ensembled deep learning model but had strong activation and the same classification result as that of the ensemble deep learning model (with reddish regions corresponding to more attention in the heatmap on each row). a** The image from an infant with BA diagnosed correctly by the model, and the region of interest was consistent between the model and human experts. **b** The image from an infant with BA diagnosed correctly by the model, and the region of interest was inconsistent between the model and human experts. **c** The image from an infant with non-BA diagnosed incorrectly by the model, and the region of interest was consistent between the model and human experts. BA biliary atresia.

gallbladder images. The EDLM outperformed human experts on both internal and external validation cohorts. Moreover, combined with the prediction of the EDLM, the sensitivities of human experts in identifying patients with BA were substantially improved from 69.5–87.3 to 93.2–95.8% in image-level diagnosis and were even better than that of the EDLM alone. A higher sensitivity would lead to fewer missed diagnosis and hence benefit patients with suspected BA in clinical practice. Hence, all these findings indicate that the EDLM could not only be used to help diagnose BA in primary hospitals lacking experts but also help experienced experts to further improve their performances in the diagnosis of BA.

We also evaluated the EDLM with gallbladder photos taken by a smartphone. Although the image quality of smartphone photos was inevitably downgraded compared to the original clean images, surprisingly, the model still performed well, with similar accuracy but higher sensitivity than the experts. Another prospective study with the developed and released smartphone app showed similar diagnostic performance. This opens an opportunity of remote and convenient online diagnosis especially for rural and underdeveloped regions without experts. In China, sonographic machines for medical diagnosis are usually not allowed to connect to the internet. However, the expert-level performance of the EDLM on the smartphone app, together with the nation-wide mobile networks and low-price smartphones, would make it easy and convenient for clinical staff even in remote underdeveloped areas to upload gallbladder photos with smartphones for online and real-time diagnosis consultancy. Such photo-based online consultancy would largely improve the diagnostic accuracy particularly for those radiologists with less experience or from underdeveloped regions.

In addition, an initial video-based intelligent diagnosis showed that the EDLM, together with automatic selection of relevant images and localization of gallbladder regions, could yield similar diagnostic performance compared to those of human experts. Such video-based diagnosis avoids the manual effort in image selection and gallbladder region localization by radiologists and can be potentially embedded into the existing diagnostic US system for fully automated diagnosis of BA during medical examination.

The initial attempt to interpret the model's predictions showed that the model also attended to the gallbladder regions during diagnosis as human experts did. However, among a small proportion of the correct diagnoses, the model made decisions just based on the visual features outside the gallbladder regions. There was 0.5% of inconsistency among correctly diagnosed external validation images, which indicates that the model can make a correct diagnosis by recognizing features other than gallbladders in some circumstances. This suggests that there might exist certain non-gallbladder features associated with BA. More investigation is necessary to explore the potentially novel biomarkers for the diagnosis of BA.

Deep learning models are usually powered by a large scale of dataset[34]. However, BA is a rare disease with low incidence, making it challenging to obtain large dataset as for other diseases[27–30]. To alleviate the potential over-fitting issue due to limited training dataset, we applied a few numbers of effective strategies for model training, including the ensemble learning, data augmentation, class weight for the imbalanced dataset between the BA and the non-BA classes, dropout of neurons during learning, and transfer learning from a pre-trained deep learning model based on large-scale natural images. Experiments showed that these strategies largely improved the generalizability of the deep learning model particularly when evaluated on the external validation dataset, suggesting that such strategies may be adopted in prospective studies relevant to medical image classification.

In most of this study, the gallbladder region in each image needs to be manually located with a form of bounding box provided by radiologists, which would inevitably increase burden on human experts during diagnosis. This issue could be avoided by automatically detecting the region of gallbladder from each image, which is feasible based on the recently developed deep learning models like Faster R-CNN[35] and will be part of the future work. Furthermore, the initial investigation of model interpretation told us that, if there were vascular or intestinal gas interference around the gallbladder, the model might mistakenly identify these interfering tissues as gallbladder and made a diagnosis partly based on these non-gallbladder regions. Automatic precise localization of gallbladder could make the AI model focus on the correct gallbladder region and therefore potentially further improve the performance of intelligent diagnosis. This may be achieved by the recently developed deep learning-based semantic image segmentation models like the U-Net[36] and DeepLab[32]. The more automatic precise localization of gallbladder regions would also enable more accurate video-based intelligent diagnosis. In addition, recent study[37] showed that the AI performance could be improved when using three-dimensional sonographic data. Therefore, one possible future work is to use sonographic volume data to potentially further improve the performance of the deep learning model.

In conclusion, we developed an EDLM that outperforms human experts in the diagnosis of BA based on a set of relatively small-scale sonographic gallbladder images acquired from five different hospitals. The generalization capability of the model was confirmed with an external validation dataset obtained from another six hospitals. Moreover, this model is potentially deployable in multiple application scenarios, such as remote diagnosis based on a smartphone app to conveniently help the unexperienced radiologists in primary hospitals, diagnosis based on the combined predictions of the model and human radiologists to further improve the diagnosis sensitivity even for experienced radiologists in tertiary hospitals. To the best of our knowledge, this is the first deep learning model for the diagnosis of BA based on sonographic gallbladder images. Since there are still lots of underdeveloped regions without sufficient healthcare support and experts for diagnosing BA all over the world, the application of the EDLM in clinical practice will benefit those jaundiced infants with suspected BA.

## Methods

**Patients and data collection**. This multicenter study was approved by the institutional Clinical Research Ethics Committee of the First Affiliated Hospital of Sun Yat-sen University, and written informed parental consent was obtained before collecting the sonographic images from each patient. Prospective research of this study was also registered at www.chictr.org.cn (ChiCTR1800017428).

Infants age <5 months with hyperbilirubinemia (serum direct bilirubin level >17.1 µmol/L and the ratio of direct to total bilirubin level >20%)[38] and suspected of BA were initially selected from 11 hospitals (Supplementary Note 1) between January 2010 and June 2019 (Fig. 7). The exclusion criteria for patients were as follows: (1) the final diagnosis was unclear; (2) jaundice was caused by bile duct obstruction to which abdominal mass compression gave rise; (3) the patient had a history of abdominal surgery; and (4) the visualization of gallbladder was indeterminate. For the patients satisfying (4), the diagnosis was highly suggestive of BA and referral to the experienced centers for further examination would be recommended. In order to expand the sample size, we also randomly selected some infants from the same 11 hospitals who did not have any known liver diseases and were considered as non-BA with a normal transcutaneous bilirubinometer test. Finally, a total of 1100 patients with suspected BA and 339 infants without jaundice were enrolled. Of the 1100 patients with suspected BA, 432 infants had BA and 668 infants had non-BA. All diagnoses were confirmed by intraoperative cholangiography under laparoscopy, percutaneous US-guided cholecystocholangiography, liver biopsy, or follow-up. The demographic characteristics and serum bilirubin level of all the included infants are listed in Supplementary Table 4.

All images were reviewed by a senior sonography expert (L.Z.) and those in poor quality were excluded. We finally retrospectively and prospectively obtained

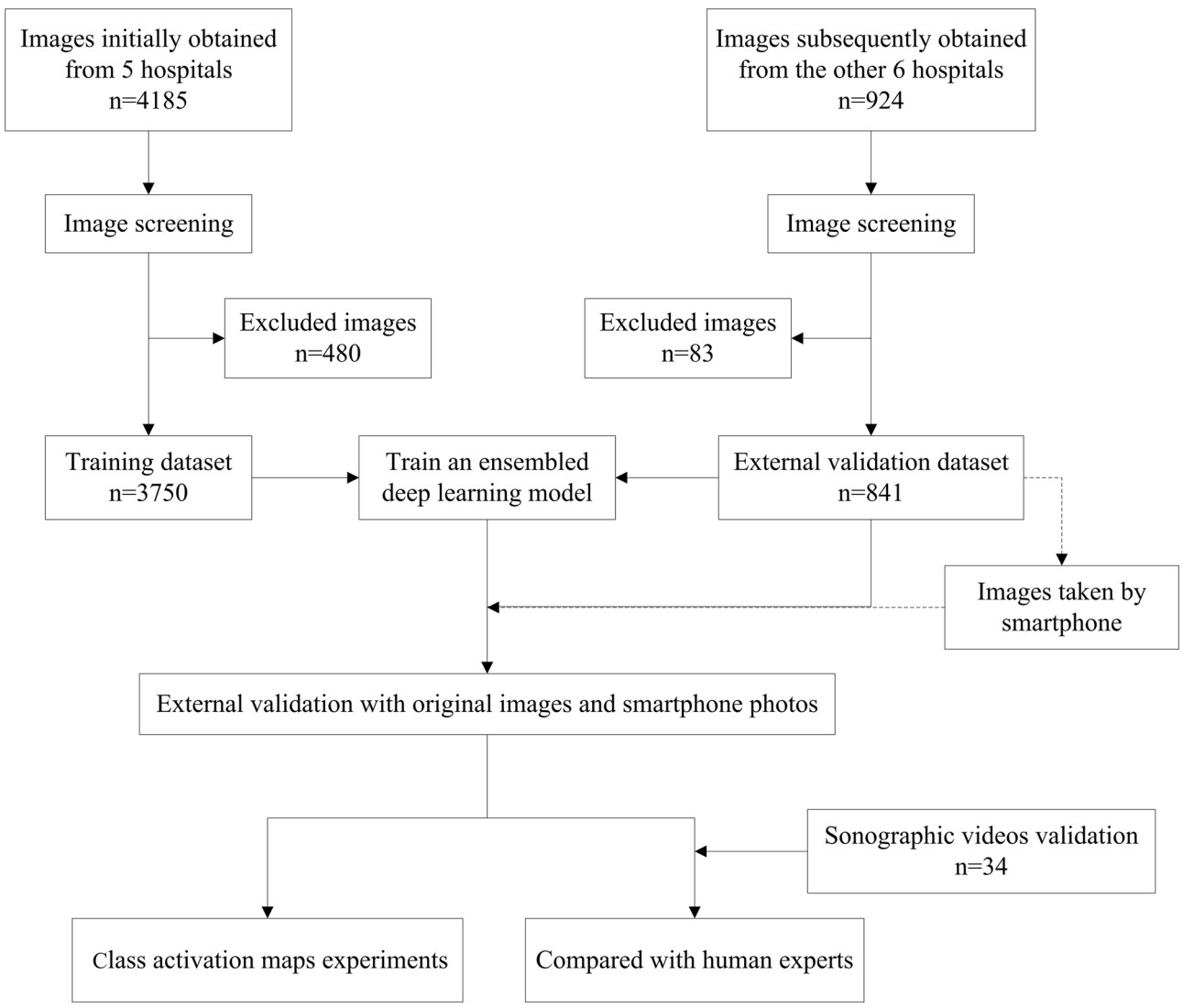

**Fig. 7 Flow chart of the study.** For each infant, sonographic gallbladder images were acquired either prospectively or retrospectively. The prospective image acquisition needed to satisfy the following criteria: (1) images were acquired after the patient fasted for at least 2 h; (2) gallbladder was detected by high-frequency transducers (>7 MHz); (3) a complete outline of the gallbladder long axis was included; (4) there was no mark or caliper within the image; (5) the depth of the image was <5 cm; (6) the image resolution was large enough (often larger than 300-by-300 pixels); (7) at least 2 independent gallbladder images were obtained from each patient. When images were acquired retrospectively, at least the criteria (1), (2), (3), (4), and (6) need to be satisfied.

3705 sonographic gallbladder images (925 from 330 patients with BA, 2780 from 811 patients without BA) from the principal hospital and 4 collaborating hospitals as the training cohort and prospectively obtained 841 sonographic gallbladder images (236 images from 102 patients with BA and the other 605 images from 196 patients without BA) from the remaining 6 collaborating hospitals as external validation cohort (Supplementary Table 5). Considering that each image included irrelevant regions (e.g., dark regions close to image boundaries and text information around the top regions), a bounding box containing the entire gallbladder was manually drawn with the free software ImageJ (version 1.52a) by two radiologists (W.Z. and W.G.), and then a senior doctor (L.Z.) double-checked and ensured that the bounding box was selected appropriately.

**Diagnosis by human experts**. In order to evaluate the efficacy of the deep learning approach, the performance of human experts was obtained in advance for direct comparison between the AI model and humans. To obtain the patient-level diagnostic performance, each random-ordered patient's image data in the training cohort was presented and diagnosed as either BA or non-BA independently by each of the two human experts (J.L. and C.Y.), and each patient's image data in the external validation cohort was presented and diagnosed independently respectively by the other three human experts (Z.W., D.C. and X.D.), both only based on all the available (often 1–3) images for each patient diagnosis. All five experts had >10 years of experience with pediatric US. Similarly, to obtain the image-level diagnosis performance, each image without any patient ID information was presented

randomly and diagnosed independently by the same experts as for the patient-level diagnosis. All these five experts have not read any of the patient images before attending this study and had no access to any other patient information (e.g., clinical history, other imaging results, etc.) during their diagnoses.

**Ensembled deep learning framework**. In this study, two types of effective AI techniques called deep CNNs and ensemble learning were adopted and combined together for intelligent diagnosis of BA. Multiple (e.g., 5 here) CNNs were trained with the training cohort and then the output predictions of these CNNs were averaged to predict the class label of each image in either the internal or the external validation dataset, resulting in an EDLM (Fig. 8a). Specifically, the training cohort was randomly separated into five complementary subsets (i.e., fivefolds), each containing the images of an equivalent number of patients. Then, each CNN was trained with four subsets and the training was stopped when the performance of the CNN started to decrease on the remaining subset. The subset used to determine the time point to stop the training of each CNN was unique (e.g., subset 1 for first CNN, and subset 2 for second CNN), which also means that the combination of four subsets for training each CNN was also unique (e.g., subset 2–5 for first CNN, and subsets 1, 3–5 for second CNN). In this way, we not only solved the issue about when to stop training a CNN but also made the five trained CNNs a bit more diverse from each other, where the diversity among CNNs would improve the generalization ability of the ensembled model as confirmed in the empirical evaluation. The adopted CNN model Se-ResNet (Supplementary Fig. 4) and

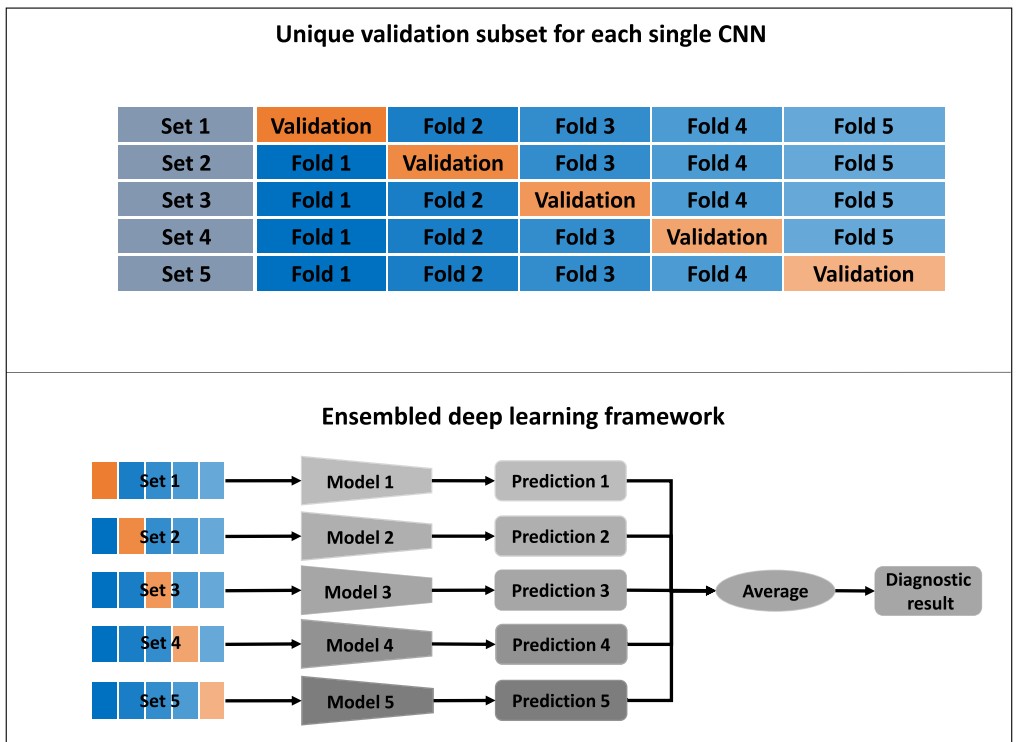

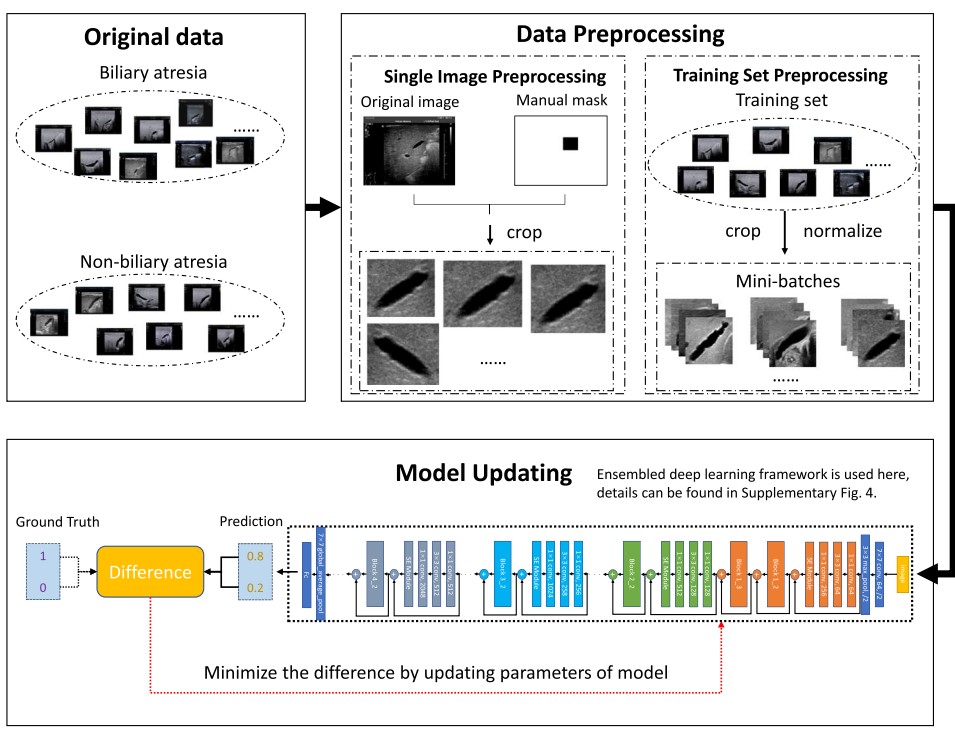

**Fig. 8 The ensembled deep learning approach for this study. a** The ensembled deep learning framework. **b** The training process for each individual CNN model. CNN convolutional neural network.

the training of each Se-ResNet (Fig. 8b) are described in detail in Supplementary Notes 2 and 3. Note that, although the training cohort was the whole internal dataset (3705 images) for the external validation, the training cohort for each of the five internal cross-validation datasets was the remaining four subsets. That means, for the internal validation, every time four of the subsets were used as a training cohort (which is divided into five new subsets) to train an ensembled deep learning model, and the ensembled model was then applied to predict the category of each image in the remaining one (internal validation) subset. Such a process was repeated five times, each time using a unique subset as the internal validation dataset.

Comparing with the existing ensemble strategies using all training data for each individual model, the proposed ensemble strategy was better than existing ones, which was supported with different CNN backbones (Supplementary Note 4 and Supplementary Tables 6–9).

**Measurements of the diagnostic performance**. The performance of each EDLM was evaluated on the test (validation) dataset, with the test dataset varied for different purposes (as seen in the "Results" section). By comparing the predicted classes from the model with the ground-truth classes obtained in advance over all the test images, the sensitivity, specificity, accuracy, positive predictive value, negative predictive value, and AUC of the ensembled model were calculated. The confidence intervals for sensitivity and specificity were calculated using the "exact" Clopper–Pearson confidence interval. At the image level, the ROC curve of the EDLM was generated by varying the threshold for the output prediction of the model, where the threshold was used to binarize the model's real-number output. Different thresholds could lead to different binary predictions of the model for each image and therefore resulted in different sensitivities and specificities on the test dataset. Similarly, at the patient level, a specific threshold would lead to the specific binary predictions for the (often multiple) images of each patient and therefore resulted in one specific binary prediction for each patient after the majority voting over the multiple binary predictions of the images from the same patient. Then, by varying the thresholds, one ROC curve would be generated based on the sequence of sensitivities and specificities at the patient level. The confidence interval for the AUC was calculated using the Binomial exact confidence interval. In addition, for comparison, the above measures were also obtained for human experts based on their diagnostic results and the ground-truth classes for the test images.

**Statistical analysis**. Differences between various AUCs were compared using Delong test. The agreement between human experts was assessed by weighted $\kappa$ statistics. The agreement was graded as follows: poor ($\kappa < 0.20$), moderate ($\kappa = 0.20$–$<0.40$), fair ($\kappa = 0.40$–$<0.60$), good ($\kappa = 0.60$–$<0.80$), or very good ($\kappa = 0.80$–$1.00$).

All statistical tests were two sided and $P$ values $< 0.05$ indicated statistically significant differences. The analyses were performed with the SPSS software package version 25 (IBM Corporation, Armonk, NY) and MedCalc Statistical Software version 15.2.2 (MedCalc Software bvba, Ostend, Belgium). All the required libraries for training the model are available in Supplementary Note 5.

**Statistics and reproducibility**. The EDLM was verified and replicated using regular machine learning metrics on external validation dataset. The software of the model was released for evaluation on new data.

**Reporting summary**. Further information on research design is available in the Nature Research Reporting Summary linked to this article.

## Data availability
Excel files containing raw data for Figs. 1, 2, and 3c and Tables 1 and 2 can be found in Supplementary Materials. Compressed images from the training dataset and external validation dataset are available at https://zenodo.org/record/4445734[39]. All other datasets generated and analyzed in the current study (including original image data) are available from the corresponding author (L.Z.) on reasonable request. Source data are provided with this paper.

## Code availability
The training code base for the deep learning framework is available at: https://github.com/youngyzzZ/Sonographic-Gallbladder-Images-for-BA-Diagnosis[39].

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

## Acknowledgements

We thank Professor Yunchao Chen from Xiang'an Hospital of Xiamen University and Dr. Qi Yu from Sanya City Womenfolk and Infant Health Care Hospital for providing data to support this study. This work was supported by National Natural Science Foundation of China (Nos.: 81501480, 81530055, 62071502, and U1811461), National Key Research and Development Program (No.: 2018YFC1315402), Guangdong Basic and Applied Basic Research Foundation (No.: 2019A1515010549), and Guangdong Key Research and Development Program (Nos.: 2019B020228001 and 2020B1111190001).

## Author contributions

W.Z. and Y.Y., image processing, performing experiments, data analysis, drafting; C.Y., J.L., X.D., Z.W., and D.C., providing images and image diagnosis; Q.L., F.Q., J.Z., and H.J., providing images; Z.L., manuscript revising; W.G.; image processing; X.X., R.W., and L.Z., study design, performing experiments, data analysis, drafting, accountability for all aspects of the work. All authors have seen and approved the final version of the manuscript.

## Competing interests

The authors declare no competing interests.

## Additional information



