## [Peer Review File · Nature Communications]

Reviewers' Comments:

Reviewer #1:

Remarks to the Author:

The manuscript presents a deep learning based approach to biliary atresia diagnosis in ultrasound images. While the manuscript presents a solution to an important medical problem, I believe that the study lacks novelty and that the results have not been assessed accordingly. Please, find my specific comments below:

- there is little to no novelty in respect to machine learning methods. The authors used well known network architecture and simple transfer learning method to develop the models. The same stands for the methods used to combine predictions of the experts and the models.

- "In this way, we not only solved the issue about when to stop training a CNN, but also make the five trained CNNs a bit more diverse from each other, where the diversity among CNNs would improve the generalization ability of the ensembled model as confirmed in the empirical evaluation." But this was not quantitatively examined. Perhaps you would obtain better results with an ensemble to 10 models? Moreover, each model forming the ensemble was based on the same network pre-trained on the ImageNet dataset. I would hardly expect any meaningful differences between the models forming the ensemble.

- "Second, a pre-trained Se-ResNet based on the large-scale natural image dataset ImageNet was used to initialize the model parameters." ImageNet images are RGB, while US images are gray-scale. How did you adjust the pre-trained model to work with grayscale images?

- there is no significance testing. Please, use statistical testing to convince readers about the usefulness of the proposed method.

- please, provide standard deviations for the performance scores (e.g. AUC).

- what about the agreement between the experts? Was it high/low?

- "Specially, 12 images of BA were misdiagnosed as non-BA by all three experts but were accurately identified by the EDLM." What about the opposite? Were there any cases mis-classified by the algorithm, but correctly assessed by the experts?

- "Diagnosis based on smartphone photos of sonographic images by the EDLM." I hardly can believe that such system would work in practice. There are many questions that put into question the usefulness of such approach, which the authors partially list (e.g. lighting environment). First, US images displayed by scanners commonly include annotations. Second, classification performance may depend on resolution of smartphone images. "sonographic machines for medical diagnosis are usually not allowed to connect to the internet." but is it difficult to upload the images directly to a smartphone?

- figure S4. 7x7 average_pool -> Fc 2. It might be good to indicate that the global average pooling was applied.

- I am not sure if I understand the CAM experiments correctly. Originally, the CAM technique was developed for multi-label classification, but the authors used it for binary classification. Figure S4 suggests there were two activation maps, for the BA and non-BA images (related to each other). Could you please clarify? What stands for the blue and red color regions in the activation maps in Fig. 6? Did the authors considered the bias term in the fc 2 layer somehow during map generation? I think the averaged activation map should be related to the output of the model (prediction probability), therefore the blue regions might be also important for the decision process. Seems, according to Fig. 6c, that the network also considered the region above the

gallbladder. Please, also provide information in Fig. 6 about which images present BA.

- for an model ensemble, did you average activation maps obtained for each model to yield the final activation map?

- "98.1% were consistent in decision making between the EDLM and human experts." Does it mean the model and the experts had both to highlight with red the gallbladder region?

Reviewer #2:

Remarks to the Author:

This is a very interesting and useful deep learning application for diagnosis of Biliary Atresia from Sonographic Gallbladder Images.

The authors also propose a smartphone version to make it possible its use without internet connection in "... more medical centers particularly from rural areas".

They propose the architecture, called "Ensembled Deep Learning Model" displayed in Figure 8, is composed by 5 CNN, trained with five different data sets and the results are averaged to produce the final binary classification.

An automatic method is used to identify the gall bladder localization in each frame using "the well-known semantic image segmentation model DeepLab"

It is mainly an application paper with a strong experimental component where the acquisition and validation of a large number of images is the central challenge. (Figure 7). A significant number of images (3705 sonographic gallbladder images - 925 from 330 patients with BA and 2780 from 811 patients without BA) was used.

This work is an excellent exemple of image processing tools applied to medical diagnosis. From a methodological point of view, however, the paper is of limited novelty and sophistication. In particular the option of using five CNN, trained by different data sets, with images randomly select from the main dataset, and averaging the results is questionable and need to be compared with other obvious strategies.

This paper is more appropriated to journals on radiology, ultrasound or dedicated to computer added diagnosis tools from imaging.

Consider the following topics that could be clarified and improved:

1. The authors compare the automatic results with the diagnosis of three human experts and conclude the former outperforms the experts. However, three experts is a very small number for a this type of study. In fact in line 131 the authors refer only two experts. This detail should be clarified.

2. The idea of conjugating the automatic classification with the classification of the expert is very interesting but in practice this corresponds to the normal strategy of any classical CAD (Computer aided Diagnosis) system. What the authors prove is that the proposed system is able to help the expert to improve the diagnosis.

3. The architecture of the system is not clearly motivated. In fact they claim that "In this way, we not only solved the issue about when to stop training a CNN, but also make the five trained CNNs a bit more diverse from each other, where the diversity among CNNs would improve the generalization ability of the ensembled model as confirmed in the empirical evaluation." The

authors do not show that this strategy is better than the obvious solution of using a single CNN trained with the entire data set. The stop criterion is not a good reason to use this architecture.

4. "The ROC curves of the model at both levels also confirmed its superior performance over human experts (Figure 1)".

How are the ROC curves displayed in Fig.1, 2 and 3 and referred in line 135 computed?

5. Line: 162: "Furthermore, when using images of mediocre quality (with frequency < 14MHz, scanning period ≤ 2018 , supersonic + others or Mindary + others) ". However in line 420 it is suggest you are using "high-frequency transducers (> 7 MHz) " which may suggest you may are using images of mediocre quality. Is that right?

6. Line 181: "Another way to obtain patient-level performance is from the diagnosis of a single image for each patient, where the single image was chosen from the multiple images of the patient by a radiologist based on the imaging quality ". Why not using an objective metric to assess image quality?

7. Table 2: It should be clear that C, D, and E refer to experts. Why not A, B and C?

8. Line 207: "One possible way to achieve such a goal is to combine the diagnosis of human expert with that of the deep learning model. ", How is this combination performed?

9. Fig 5 is not clear, and it is not described in detail in the text. The first block is supposed to be the gall bladder segmentation procedure, but none of its sub-blocks are identified.

Reviewer #3:

Remarks to the Author:

The study goes far beyond my skills which are limited to the knowledge of clinical and diagnostic issues rather than to the Ensembled Deep Learning Models. I do not doubt that the scientific methodology of the study is rigorous and correct but I would like more clinical, laboratory, genetic information on patients with biliary atresia and non-biliary atresia patients included in the study. The diagnosis of biliary atresia is difficult because it is difficult to distinguish biliary atresia patients from patients with intrahepatic forms of cholestasis that tend to present similar clinical, sonographic, laboratory and histological pictures.

Thank you for the very valuable comments and suggestions! We have performed additional experiments and analysis as suggested, and revised the manuscript accordingly, with main revisions highlighted. We also revised the manuscript according to the formatting requirements of your journal of *Nature communications*, with main revisions highlighted. In addition, we found there was a mistake in the original codes that produced the CAM when we were checking the whole manuscript. So, we replaced the Fig.6 with right ones and corrected related data.

The detailed response to each comment is listed below **in red**.

Reviewer #1:

1) *“there is little to no novelty in respect to machine learning methods. The authors used well known network architecture and simple transfer learning method to develop the models. The same stands for the methods used to combine predictions of the experts and the models.”*

Response: We would like to emphasize that the main objective of this study is not to develop a novel deep learning approach but to apply deep learning technique to an important medical problem. The expert-level AI model developed in this study and the combined human-model diagnosis would help radiologists largely improve their diagnosis performance particularly in undeveloped regions. On the other hand, although the proposed ensemble model is based on the existing ensemble strategy, the (slight) novelty of the proposed ensemble strategy is to use part of the training dataset to train an individual model each time and validate with the remaining part, while existing ensemble strategy uses all training data for each individual model. Our comprehensive empirical comparisons (Supplementary Tables 6 to 9) showed that the proposed ensemble strategy is better than existing ones.

2) *“In this way, we not only solved the issue about when to stop training a CNN, but also make the five trained CNNs a bit more diverse from each other, where the diversity among CNNs would improve the generalization ability of the ensemble model as confirmed in the empirical evaluation.” But this was not quantitatively examined. Perhaps you would obtain better results with an ensemble to 10 models? Moreover, each model forming the ensemble was based on the same network pre-trained on the ImageNet dataset. I would hardly expect any meaningful differences between the models forming the ensemble.”*

Response: Thank you for the comments! We are sorry we did not show the empirical comparison in the first submission. Based on the comments, we added a series of empirical comparisons, including the results from (1) an individual CNN model trained with the entire training dataset, (2) the general ensemble model (of 5 and 10 individual models respectively) where each individual model is trained with the entire training dataset, and (3) different CNN backbones, all showing superior performance

from the proposed ensemble strategy (Supplementary Tables 6 to 9). In our ensemble strategy, multiple individual models starting from the same pretrained network would result in different model parameters (therefore different individual models) because each individual model was trained with a different but partially overlapped subset of the training dataset. With such different individual models, the ensemble model would often perform better than single models. Actually, even using the same set of data to train multiple individual models, due to randomness of mini-batches during model training, all these individual models would result in different model parameters, which also would be able to form an ensemble model better than individual ones. This has been confirmed in both previous studies with natural images and in our study here. Last but not least, the proposed ensemble strategy is expected to be directly applied to general intelligent diagnosis tasks, and we would release our source code once the study is ready to be published.

3) *“Second, a pre-trained Se-ResNet based on the large-scale natural image dataset ImageNet was used to initialize the model parameters.” ImageNet images are RGB, while US images are gray-scale. How did you adjust the pre-trained model to work with grayscale images?”*

Response: Each single-channel grayscale image become a three-channel image by simply duplicating the original single channel three times. It is worth noting that this is a general routine for a pretrained AI model to process grayscale images.

4) *“there is no significance testing. Please, use statistical testing to convince readers about the usefulness of the proposed method.”*

Response: As suggested, we have added the significance testing information where necessary in the new manuscript.

5) *“please, provide standard deviations for the performance scores (e.g. AUC).”*

Response: As suggested, we have added the standard deviation for the performance scores.

6) *“what about the agreement between the experts? Was it high/low?”*

Response: We calculated the agreement between the three experts for external validation and the two experts for internal validation. Overall the agreement is around 0.306 to 0.683. The information has been added in the new manuscript.

7) *“Specially, 12 images of BA were misdiagnosed as non-BA by all three experts but were accurately identified by the EDLM.” What about the opposite? Were there any cases mis-classified by the algorithm, but correctly assessed by the experts?”*

Response: Yes. On the external validation dataset, there were 20 images misdiagnosed by three experts, but correctly diagnosed by the model. Of the 20 images, 12 images were false negatives (BA misdiagnosed as non-BA), and 8 images were false positives (non-BA misdiagnosed as BA). On the contrary, there were also 20 images misdiagnosed by the model, but correctly assessed by three experts on the external

validation dataset, of which 14 images were false positives and 6 images were false negatives. Overall, experts missed more BA cases than the AI model. Such detailed information has been added in the new manuscript.

8) *“Diagnosis based on smartphone photos of sonographic images by the EDLM.” I hardly can believe that such system would work in practice. There are many questions that put into question the usefulness of such approach, which the authors partially list (e.g. lighting environment). First, US images displayed by scanners commonly include annotations. Second, classification performance may depend on resolution of smartphone images. “sonographic machines for medical diagnosis are usually not allowed to connect to the internet.” but is it difficult to upload the images directly to a smartphone?”*

Response: Thanks for your concerns.

For the first concern about annotations on US images, this issue does not exist because doctors are instructed to just take photo of the gallbladder region from the machine screen (i.e., excluding other structures and the surrounding annotation information), such that only the US image regions covering the gallbladder region was included in the smartphone photos. This can be done by keeping the smartphone camera close to the screen of US scanners during photo taking.

For the second concern, classification performance probably does not depend on the resolution of smartphone images, because the resolution (larger than 1000x1000 pixels) of smartphone images is actually larger than that of the US images (often less than 700x700 pixels) used for AI model training. The relatively higher-resolution smartphone images were resized (downscaled) before being fed to the AI model.

For the third concern, each original US image was not ‘uploaded’ to a smartphone. Instead the smartphone takes photo of the US image from the scanner’s screen. Since the smartphone app includes the function of photo taking, users just need to open the smartphone app and directly take photos following the simple instruction inside the app, after which the photo will be directly transferred to the remotely located AI system for intelligent diagnosis.

Actually, the smartphone app has been released to a small group of users in several different hospitals and the initial evaluation based on smartphone app is positive, i.e., the performance of the smartphone app is close to human experts as reported in the manuscript.

9) *“figure S4. 7x7 average_pool -> Fc 2. It might be good to indicate that the global average pooling was applied.”*

Response: The ‘average_pool’ was replaced by the ‘global_average_pool’.

10) *“I am not sure if I understand the CAM experiments correctly. Originally, the CAM technique was developed for multi-label classification, but the authors used it for binary classification. Figure S4 suggests there were two activation maps, for the BA and non-BA images (related to each other). Could you please clarify? What stands for the blue and red color regions in the activation maps in Fig. 6? Did the authors*

considered the bias term in the fc 2 layer somehow during map generation? I think the averaged activation map should be related to the output of the model (prediction probability), therefore the blue regions might be also important for the decision process. Seems, according to Fig. 6c, that the network also considered the region above the gallbladder. Please, also provide information in Fig. 6 about which images present BA.”

Response: First we feel sorry because we made a mistake in the CAM analysis. We found one bug in the original codes that produced the CAM. Therefore, we replaced Fig.6 with right ones and corrected related data.

Yes, here CAM is used for a binary classifier, therefore two activation maps can be generated by CAM for each input image. The blue to red colors stand for averaged activation strengths (from low to high values, color bars added in the new manuscript). In particular, dark blue represents that the activation is zero. Since the activation strength indicates the importance of the regions to the final output, the blue regions contribute little to the final prediction (of the output neuron with higher probability). In Figure 6c, the classifier diagnosed incorrectly, and it is possible for the classifier to attend to regions besides the gallbladder particularly for incorrect diagnosis.

The bias term in the FC layer was not considered in the CAM method (because CAM method basically computes the importance of each kernel’s output, also called feature map, and then obtain the weighted sum of all feature maps in the last convolutional layer). In addition, as suggested, we have provided information about which images present BA in the updated figure caption.

11) *“for an model ensemble, did you average activation maps obtained for each model to yield the final activation map?”*

Response: No, an activation map was generated based on one single model rather than the ensembled model, because different single models in an ensemble model may have different prediction results, which would result in more or less different activation maps. The single model which has strong activation and the same prediction result as the ensembled model was chosen to generate the activation map, and the activation map is representative enough to show the attended region during ensemble model prediction.

12) *“98.1% were consistent in decision making between the EDLM and human experts.” Does it mean the model and the experts had both to highlight with red the gallbladder region?”*

Response: The ‘consistent in decision making’ means that the attended region by the model (obtained by the CAM method) covers or partly covers the (gallbladder) regions used by human experts for diagnosis. Note that human experts did not need to highlight with red the gallbladder region.

Reviewer #2:

1) *“This work is an excellent example of image processing tools applied to medical diagnosis. From a methodological point of view, however, the paper is of limited novelty and sophistication. In particular the option of using five CNN, trained by different data sets, with images randomly select from the main dataset, and averaging the results is questionable and need to be compared with other obvious strategies.”*

Response: We agree that our study is application-oriented. However, we believe the proposed ensemble deep learning strategy is generalizable and better than state-of-the-art strategies. To support this claim, we have performed a series of experiments by considering (1) a single CNN model trained with the entire training dataset, (2) the general ensemble strategy where each single model is trained with the entire training dataset, and (3) different CNN backbones, all showing superior performance from the proposed ensemble strategy (Supplementary material Table 6 to 9). What’s more, the implementation is not sophisticated, and we plan to release our source code once the study is ready to be published. The proposed ensemble strategy is expected to be directly applied to general intelligent diagnosis tasks.

2) *“The authors compare the automatic results with the diagnosis of three human experts and conclude the former outperforms the experts. However, three experts is a very small number for a this type of study. In fact in line 131 the authors refer only two experts. This detail should be clarified.”*

Response: In general, more experts would help prove the effectiveness of an AI model in studies. However, there is a lack of experts in this study because biliary atresia is a rare disease with 1/5000-1/10000 of infants in China. Six experts (one expert screened all images for quality control and was not involved in diagnosis study) in our study were almost half of all experts who are good at Ultrasonic diagnosis of biliary atresia in the South and West of China. Because of the shortage of experts in this field, diagnosis of many infants with biliary atresia are often delayed. This is also the reason why we launched the study.

In this study, because two of the involved experts provided part of external validation images, they were asked to evaluate the training dataset independently only. Similarly, because of the other three experts provided most of training images, they were asked to evaluate the external validation images independently.

3) *“The idea of conjugating the automatic classification with the classification of the expert is very interesting but in practice this corresponds to the normal strategy of any classical CAD (Computer aided Diagnosis) system. What the authors prove is that the proposed system is able to help the expert to improve the diagnosis.”*

Response: Yes! One purpose of this study is to build a deep learning model to help experts/doctors improve their diagnosis of biliary atresia.

4) *“The architecture of the system is not clearly motivated. In fact they claim that “In this way, we not only solved the issue about when to stop training a CNN, but also make the five trained CNNs a bit more diverse from each other, where the diversity among CNNs would improve the generalization ability of the ensembled model as*

confirmed in the empirical evaluation.” The authors do not show that this strategy is better than the obvious solution of using a single CNN trained with the entire data set. The stop criterion is not a good reason to use this architecture.”

Response: As responded to comment 1), we have performed a series of experiments to compare with existing strategies, all showing the proposed ensemble strategy is superior to existing ones. Stop criterion within the ensemble strategy is just a byproduct.

5) *“The ROC curves of the model at both levels also confirmed its superior performance over human experts (Figure 1)”. How are the ROC curves displayed in Fig.1, 2 and 3 and referred in line 135 computed?*

Response: Basically, each ROC was generated by varying the threshold for the output of the model. More details were in the “Materials and Methods: Measurements of the diagnostic performance” section.

6) *“Line: 162: “Furthermore, when using images of mediocre quality (with frequency < 14MHz, scanning period ≤ 2018, supersonic + others or Mindary + others) ”. However in line 420 it is suggest you are using “high-frequency transducers (> 7 MHz) ” which may suggest you may are using images of mediocre quality. Is that right?”*

Response: Not really. In general, transducers with higher frequency would produce images in higher quality. However, the quality of images can be influenced by other parameters as well. Therefore, the quality of images was not merely determined by the frequency of transducers. Actually, all images included in the study were qualified for further analysis after an expert screened all images and excluded unqualified images. The aim of dividing training data into different groups for further analysis was to evaluate generalization of the model. Although the quality of images with frequency $\geq 14\text{MHz}$ is better than that with frequency $< 14\text{MHz}$, it did not mean the images with frequency $< 14\text{MHz}$ is mediocre. We noticed that the word ‘mediocre’ could be inappropriate and misleading and therefore has been discarded in the new manuscript (instead, using words like ‘images in relatively moderate quality’).

7) *“Line 181: “Another way to obtain patient-level performance is from the diagnosis of a single image for each patient, where the single image was chosen from the multiple images of the patient by a radiologist based on the imaging quality ”. Why not using an objective metric to assess image quality?”*

Response: The radiologist who chose the single image for each patient did use an objective metric, i.e., choosing the images with clear presentation of the contour of gallbladder. We have added this in the new manuscript. Such objective metric could be implemented by computer programming (as part of future work).

8) *“Table 2: It should be clear that C, D, and E refer to experts. Why not A, B and*

C?”

Response: As suggested, ‘Expert C/D/E’ replaced the original ‘C/D/E’. Different experts (C, D, E) were employed in the external validation (Table 2) who did not have any access to the training dataset. In comparison, experts A and B were involved in the internal validation (Table 1) who analyzed the training dataset. In this way, it is assured that experts involved in external validation would not be affected by any information relevant to the training dataset.

9) *“Line 207: “One possible way to achieve such a goal is to combine the diagnosis of human expert with that of the deep learning model. “, How is this combination performed?”*

Response: The combination is performed as follows. One case is diagnosed as BA (biliary atresia) if the diagnosis of the model or a human expert is BA, and is diagnosed as non-BA only if the diagnoses of both the model and the human expert are non-BA. Such combination would improve the sensitivity of diagnosis which is preferable in real clinical scenarios.

10) *“Fig 5 is not clear, and it is not described in detail in the text. The first block is supposed to be the gall bladder segmentation procedure, but none of its sub-blocks are identified.”*

Response: Yes, the first block is for gallbladder segmentation. Detailed information about each convolutional layer has been included in the figure.

Reviewer #3:

1) *“I do not doubt that the scientific methodology of the study is rigorous and correct but I would like more clinical, laboratory, genetic information on patients with biliary atresia and non-biliary atresia patients included in the study. The diagnosis of biliary atresia is difficult because it is difficult to distinguish biliary atresia patients from patients with intrahepatic forms of cholestasis that tend to present similar clinical, sonographic, laboratory and histological pictures.”*

Response: We totally agree that the diagnosis of biliary atresia is difficult because of the overlapped clinical symptoms and laboratory findings between biliary atresia and other intrahepatic cholestasis. To minimize misdiagnosis, each infant with biliary atresia was confirmed with at least one of the following procedures: surgery exploration, intraoperative cholangiography, percutaneous US-guided cholecysto-cholangiography, liver biopsy and follow-up. Each of these procedures has been widely accepted as the reference standard for biliary atresia diagnosis or biliary atresia exclusion. As suggested, demographic and laboratory information of all included infants has been added (supplementary Table 4). However, the genetic information is available only for a few subjects since the gene sequencing test is highly resource-consuming and unaffordable for many families. In addition, a vast number of health infants, who did not have any known liver diseases and therefore

had no clinical information, were included to develop the AI diagnosis system in the study.

We sincerely hope our reponse and revision will meet you the quality of publishment in your journal of NATURE COMMUNICATIONS.

Best Regards,

Luyao Zhou

Reviewers' Comments:

Reviewer #1:

Remarks to the Author:

Reviewer #1:

1) "there is little to no novelty in respect to machine learning methods. The authors used well known network architecture and simple transfer learning method to develop the models. The same stands for the methods used to combine predictions of the experts and the models."

Response: We would like to emphasize that the main objective of this study is not to develop a novel deep learning approach but to apply deep learning technique to an important medical problem. The expert-level AI model developed in this study and the combined human-model diagnosis would help radiologists largely improve their diagnosis performance particularly in undeveloped regions. On the other hand, although the proposed ensemble model is based on the existing ensemble strategy, the (slight) novelty of the proposed ensemble strategy is to use part of the training dataset to train an individual model each time and validate with the remaining part, while existing ensemble strategy uses all training data for each individual model. Our comprehensive empirical comparisons (Supplementary Tables 6 to 9) showed that the proposed ensemble strategy is better than existing ones.

Thank you very much for the response. I understand that the main aim of the work was to develop a deep learning model for a concrete medical problem. I was impressed by your multicenter study and the work of the experts who examined the data. Nevertheless, you used a simple deep learning method for classification, originating from a paper published several years ago. I don't see this as a very important issue (your method outperformed the experts after all), but given the latest developments in the field of deep learning, especially transfer learning, I believe it would be straightforward to improve your method. For example, you duplicated the US images into the RGB color channels to enable transfer learning, but it would be probably better to use a network trained on gray-scaled ImageNet images. You could also use image colorization, which would enable extraction of better performing features from the first blocks of the pre-trained model (there are no color gradients if you duplicate images). Moreover, it is unclear what transfer learning did you use (sorry if I missed this information). Did you fine-tune the entire network or perhaps several last convolutional blocks? Modern approaches to transfer learning include automatic selection of convolutional blocks to fine-tune.

2) ""In this way, we not only solved the issue about when to stop training a CNN, but also make the five trained CNNs a bit more diverse from each other, where the diversity among CNNs would improve the generalization ability of the ensembled model as confirmed in the empirical evaluation." But this was not quantitatively examined. Perhaps you would obtain better results with an ensemble to 10 models? Moreover, each model forming the ensemble was based on the same network pre-trained on the ImageNet dataset. I would hardly expect any meaningful differences between the models forming the ensemble."

Response: Thank you for the comments! We are sorry we did not show the empirical comparison in the first submission. Based on the comments, we added a series of empirical comparisons, including the results from (1) an individual CNN model trained with the entire training dataset, (2) the general ensemble model (of 5 and 10 individual models respectively) where each individual model is trained with the entire training dataset, and (3) different CNN backbones, all showing superior performance from the proposed ensemble strategy (Supplementary Tables 6 to 9). In our ensemble strategy, multiple individual models starting from the same pretrained network would result in different model parameters (therefore different individual models) because each individual model was trained with a different but partially overlapped subset of the training dataset. With such different individual models, the ensemble model would often perform better than single models. Actually, even using the same set of data to train multiple individual models, due to randomness of mini-batches during model training, all these individual models would result in different model parameters, which also would be able to form an ensemble model better than

individual ones. This has been confirmed in both previous studies with natural images and in our study here. Last but not least, the proposed ensemble strategy is expected to be directly applied to general intelligent diagnosis tasks, and we would release our source code once the study is ready to be published.

Thank you for sharing Tables 6-9, but I am a little bit confused. Do you plan to include these tables in the final paper? If yes, please provide the descriptions of the experiments in the supplementary materials.

If I understand correctly, you used the entire data for training instead of 5-fold cross-validation, which unsurprisingly resulted in worse performance. Moreover, other strategies to ensemble formation have not been investigated (as stated by the second reviewer). It would be interesting to check whether the ensemble of different deep learning models (listed in tables 6-9) leads to better classification.

Please, also clarify following issues:

- My initial concern has not been addressed. To form your ensemble, you applied the 5-fold validation, but perhaps 3-fold or 10-fold validation would work better?

- "the training was stopped when the performance of the CNN started to decrease on the remaining subset." but I could not find information how you stopped the training for the other methods (tables 6-9, training data for each individual model = All). Was there a stopping criterion, or perhaps you trained each model for 210 epochs, which the sup material suggests ('The maximum number of epochs was set to 210 before which each model has been well trained without much change in model parameters'). If there was no stopping criterion, it is not surprising that the original ensemble method performed better. Stopping based on a separate dataset commonly result in better models.

3) ""Second, a pre-trained Se-ResNet based on the large-scale natural image dataset ImageNet was used to initialize the model parameters." ImageNet images are RGB, while US images are gray-scale. How did you adjust the pre-trained model to work with grayscale images?"

Response: Each single-channel grayscale image become a three-channel image by simply duplicating the original single channel three times. It is worth noting that this is a general routine for a pretrained AI model to process grayscale images.

Thank you. Could you add this information to the manuscript?

4) "there is no significance testing. Please, use statistical testing to convince readers about the usefulness of the proposed method."

Response: As suggested, we have added the significance testing information where necessary in the new manuscript.

Thank you.

5) "please, provide standard deviations for the performance scores (e.g. AUC)."

Response: As suggested, we have added the standard deviation for the performance scores.

Thank you.

6) "what about the agreement between the experts? Was it high/low?"

Response: We calculated the agreement between the three experts for external validation and the two experts for internal validation. Overall the agreement is around 0.306 to 0.683. The information has been added in the new manuscript.

Thank you.

7) ""Specially, 12 images of BA were misdiagnosed as non-BA by all three experts but were accurately identified by the EDLM." What about the opposite? Were there any cases mis-classified by the algorithm, but correctly assessed by the experts?"

Response: Yes. On the external validation dataset, there were 20 images misdiagnosed by three experts, but correctly diagnosed by the model. Of the 20 images, 12 images were false negatives (BA misdiagnosed as non-BA), and 8 images were false positives (non-BA misdiagnosed as BA). On the contrary, there were also 20 images misdiagnosed by the model, but correctly assessed by three experts on the external validation dataset, of which 14 images were false positives and 6 images were false negatives. Overall, experts missed more BA cases than the AI model. Such detailed information has been added in the new manuscript.

Thank you.

8) ""Diagnosis based on smartphone photos of sonographic images by the EDLM." I hardly can believe that such system would work in practice. There are many questions that put into question the usefulness of such approach, which the authors partially list (e.g. lighting environment). First, US images displayed by scanners commonly include annotations. Second, classification performance may depend on resolution of smartphone images. "sonographic machines for medical diagnosis are usually not allowed to connect to the internet." but is it difficult to upload the images directly to a smartphone?"

Response: Thanks for your concerns.

For the first concern about annotations on US images, this issue does not exist because doctors are instructed to just take photo of the gallbladder region from the machine screen (i.e., excluding other structures and the surrounding annotation information), such that only the US image regions covering the gallbladder region was included in the smartphone photos. This can be done by keeping the smartphone camera close to the screen of US scanners during photo taking.

For the second concern, classification performance probably does not depend on the resolution of smartphone images, because the resolution (larger than 1000x1000 pixels) of smartphone images is actually larger than that of the US images (often less than 700x700 pixels) used for AI model training. The relatively higher-resolution smartphone images were resized (downscaled) before being fed to the AI model.

For the third concern, each original US image was not 'uploaded' to a smartphone. Instead the smartphone takes photo of the US image from the scanner's screen. Since the smartphone app includes the function of photo taking, users just need to open the smartphone app and directly take photos following the simple instruction inside the app, after which the photo will be directly transferred to the remotely located AI system for intelligent diagnosis.

Actually, the smartphone app has been released to a small group of users in several different hospitals and the initial evaluation based on smartphone app is positive, i.e., the performance of the smartphone app is close to human experts as reported in the manuscript.

Thank you.

9) "figure S4. 7x7 average_pool -> Fc 2. It might be good to indicate that the global average pooling was applied."

Response: The 'average_pool' was replaced by the 'global_average_pool'.

Thank you.

10) "I am not sure if I understand the CAM experiments correctly. Originally, the CAM technique was developed for multi-label classification, but the authors used it for binary classification. Figure S4 suggests there were two activation maps, for the BA and non-BA images (related to each other). Could you please clarify? What stands for the blue and red color regions in the activation maps in Fig. 6? Did the authors considered the bias term in the fc 2 layer somehow during map generation? I think the averaged activation map should be related to the output of the model

(prediction probability), therefore the blue regions might be also important for the decision process. Seems, according to Fig. 6c, that the network also considered the region above the gallbladder. Please, also provide information in Fig. 6 about which images present BA."

Response: First we feel sorry because we made a mistake in the CAM analysis. We found one bug in the original codes that produced the CAM. Therefore, we replaced Fig.6 with right ones and corrected related data.

Yes, here CAM is used for a binary classifier, therefore two activation maps can be generated by CAM for each input image. The blue to red colors stand for averaged activation strengths (from low to high values, color bars added in the new manuscript). In particular, dark blue represents that the activation is zero. Since the activation strength indicates the importance of the regions to the final output, the blue regions contribute little to the final prediction (of the output neuron with higher probability). In Figure 6c, the classifier diagnosed incorrectly, and it is possible for the classifier to attend to regions besides the gallbladder particularly for incorrect diagnosis. The bias term in the FC layer was not considered in the CAM method (because CAM method basically computes the importance of each kernel's output, also called feature map, and then obtain the weighted sum of all feature maps in the last convolutional layer). In addition, as suggested, we have provided information about which images present BA in the updated figure caption.

Thank you.

11) "for an model ensemble, did you average activation maps obtained for each model to yield the final activation map?"

Response: No, an activation map was generated based on one single model rather than the ensembled model, because different single models in an ensemble model may have different prediction results, which would result in more or less different activation maps. The single model which has strong activation and the same prediction result as the ensembled model was chosen to generate the activation map, and the activation map is representative enough to show the attended region during ensemble model prediction.

Thank you. Please, include this information in the manuscript.

12) "'98.1% were consistent in decision making between the EDLM and human experts." Does it mean the model and the experts had both to highlight with red the gallbladder region?"

Response: The 'consistent in decision making' means that the attended region by the model (obtained by the CAM method) covers or partly covers the (gallbladder) regions used by human experts for diagnosis. Note that human experts did not need to highlight with red the gallbladder region.

Thank you.

Reviewer #2 (response by reviewer #1):

1) "This work is an excellent example of image processing tools applied to medical diagnosis. From a methodological point of view, however, the paper is of limited novelty and sophistication. In particular the option of using five CNN, trained by different data sets, with images randomly select from the main dataset, and averaging the results is questionable and need to be compared with other obvious strategies."

Response: We agree that our study is application-oriented. However, we believe the proposed ensemble deep learning strategy is generalizable and better than state-of-the-art strategies. To support this claim, we have performed a series of experiments by considering (1) a single CNN model trained with the entire training dataset, (2) the general ensemble strategy where each single model is trained with the entire training dataset, and (3) different CNN backbones, all showing superior performance from the proposed ensemble strategy (Supplementary material Table 6 to 9). What's more, the implementation is not sophisticated, and we plan to release our

source code once the study is ready to be published. The proposed ensemble strategy is expected to be directly applied to general intelligent diagnosis tasks.

Thank you for the response. Please, see the first and the second response of the reviewer #1.

2) "The authors compare the automatic results with the diagnosis of three human experts and conclude the former outperforms the experts. However, three experts is a very small number for a this type of study. In fact in line 131 the authors refer only two experts. This detail should be clarified."

Response: In general, more experts would help prove the effectiveness of an AI model in studies. However, there is a lack of experts in this study because biliary atresia is a rare disease with 1/5000-1/10000 of infants in China. Six experts (one expert screened all images for quality control and was not involved in diagnosis study) in our study were almost half of all experts who are good at Ultrasonic diagnosis of biliary atresia in the South and West of China. Because of the shortage of experts in this field, diagnosis of many infants with biliary atresia are often delayed. This is also the reason why we launched the study.

In this study, because two of the involved experts provided part of external validation images, they were asked to evaluate the training dataset independently only. Similarly, because of the other three experts provided most of training images, they were asked to evaluate the external validation images independently.

Thank you.

3) "The idea of conjugating the automatic classification with the classification of the expert is very interesting but in practice this corresponds to the normal strategy of any classical CAD (Computer aided Diagnosis) system. What the authors prove is that the proposed system is able to help the expert to improve the diagnosis."

Response: Yes! One purpose of this study is to build a deep learning model to help experts/doctors improve their diagnosis of biliary atresia.

Thank you. Notice, however, that it would be interesting to first select the US images misdiagnosed by the experts, and train the machine learning model to pay more attention to correct classification of these cases.

4) "The architecture of the system is not clearly motivated. In fact they claim that "In this way, we not only solved the issue about when to stop training a CNN, but also make the five trained CNNs a bit more diverse from each other, where the diversity among CNNs would improve the generalization ability of the ensembled model as confirmed in the empirical evaluation." The authors do not show that this strategy is better than the obvious solution of using a single CNN trained with the entire data set. The stop criterion is not a good reason to use this architecture."

Response: As responded to comment 1), we have performed a series of experiments to compare with existing strategies, all showing the proposed ensemble strategy is superior to existing ones. Stop criterion within the ensemble strategy is just a byproduct.

Thank you for adding this analysis.

5) "The ROC curves of the model at both levels also confirmed its superior performance over human experts (Figure 1)". How are the ROC curves displayed in Fig.1, 2 and 3 and referred in line 135 computed?

Response: Basically, each ROC was generated by varying the threshold for the output of the model. More details were in the "Materials and Methods: Measurements of the diagnostic performance" section.

Thank you. But did you use trapezoidal approach for calculations? How did you compute AUCs for the experts (single point)?

6) "Line: 162: "Furthermore, when using images of mediocre quality (with frequency <14MHz, scanning period ≤ 2018 , supersonic + others or Mindary + others) ". However in line 420 it is suggest you are using "high-frequency transducers (> 7 MHz) " which may suggest you may are using images of mediocre quality. Is that right?"

Response: Not really. In general, transducers with higher frequency would produce images in higher quality. However, the quality of images can be influenced by other parameters as well. Therefore, the quality of images was not merely determined by the frequency of transducers. Actually, all images included in the study were qualified for further analysis after an expert screened all images and excluded unqualified images. The aim of dividing training data into different groups for further analysis was to evaluate generalization of the model. Although the quality of images with frequency ≥ 14 MHz is better than that with frequency <14 MHz, it did not mean the images with frequency <14 MHz is mediocre. We noticed that the word 'mediocre' could be inappropriate and misleading and therefore has been discarded in the new manuscript (instead, using words like 'images in relatively moderate quality').

Thank you. Could you also include the information about the number of cases for each subgroup in Sup Table 2?

7) "Line 181: "Another way to obtain patient-level performance is from the diagnosis of a single image for each patient, where the single image was chosen from the multiple images of the patient by a radiologist based on the imaging quality ". Why not using an objective metric to assess image quality?"

Response: The radiologist who chose the single image for each patient did use an objective metric, i.e., choosing the images with clear presentation of the contour of gallbladder. We have added this in the new manuscript. Such objective metric could be implemented by computer programming (as part of future work).

Thank you.

8) "Table 2: It should be clear that C, D, and E refer to experts. Why not A, B and C?"

Response: As suggested, 'Expert C/D/E' replaced the original 'C/D/E'. Different experts (C, D, E) were employed in the external validation (Table 2) who did not have any access to the training dataset. In comparison, experts A and B were involved in the internal validation (Table 1) who analyzed the training dataset. In this way, it is assured that experts involved in external validation would not be affected by any information relevant to the training dataset.

Thank you.

9) "Line 207: "One possible way to achieve such a goal is to combine the diagnosis of human expert with that of the deep learning model. ", How is this combination performed?"

Response: The combination is performed as follows. One case is diagnosed as BA (biliary atresia) if the diagnosis of the model or a human expert is BA, and is diagnosed as non-BA only if the diagnoses of both the model and the human expert are non-BA. Such combination would improve the sensitivity of diagnosis which is preferable in real clinical scenarios.

Thank you.

10) "Fig 5 is not clear, and it is not described in detail in the text. The first block is supposed to be the gall bladder segmentation procedure, but none of its sub-blocks are identified."

Response: Yes, the first block is for gallbladder segmentation. Detailed information about each convolutional layer has been included in the figure.

Thank you.

Additional comments:

- Please, proofread the manuscript. It includes some grammatical errors and awkward sentences.
- Fig. 8a) suggests that you averaged the training set predictions to obtain diagnostic results presented in Table 1. The performance on the training set is commonly too optimistic and should not be reported. Please, clarify.
- If I understand correctly, you did not apply the cross-validation to obtain the results presented in supp Table 1 and 2. What was the training procedure in this case?
- 'The rate of dropout was set to 0.2 for all tests in this study'. Do you mean for the training of the network? Please, specify whether you applied the dropout to all layers of the network.
- Please, better describe the transfer learning method. Clearly indicate which convolutional blocks were used for fine-tuning etc
- "The mean and standard deviation of pixel intensities were computed over all cropped images, which were then used to normalize each pixel in all rescaled images. Each normalized image will be used as an input to the SE-ResNet for model training." But the SE-ResNet paper states 'Each input image is normalised through mean RGB-channel subtraction.' Why did you apply different normalization?
- Fig. 8a), there is an issue with the prediction and set numeration.
- Fig. 8b), I would suggest to improve the resolution of the images. It is impossible to read the text, even after zooming in.
- From the sup material: 'To improve the generalizability of each SE-ResNet, besides the ensembled learning mentioned above', perhaps it would be better to replace 'above' with 'in the main text'?

Reviewer #3:

Remarks to the Author:

The revised version is much improved. As for me, it would have been useful to compare the results with a group of infants with intrahepatic cholestasis.

Reviewer #4:

None

Thank you for the very valuable comments and suggestions again! As suggested, we have performed additional experiments and the results were presented in the manuscript and partly in the response below. We also revised the manuscript (in red) accordingly.

The detailed response to each comment is listed below. Only the comments requiring further revision were included, for which all the corresponding previous responses and the previous comments were also kept for better understanding of the latest comments and the responses (in red).

Reviewer #1:

1) “there is little to no novelty in respect to machine learning methods. The authors used well known network architecture and simple transfer learning method to develop the models. The same stands for the methods used to combine predictions of the experts and the models.”

Response: We would like to emphasize that the main objective of this study is not to develop a novel deep learning approach but to apply deep learning technique to an important medical problem. The expert-level AI model developed in this study and the combined human-model diagnosis would help radiologists largely improve their diagnosis performance particularly in undeveloped regions. On the other hand, although the proposed ensemble model is based on the existing ensemble strategy, the (slight) novelty of the proposed ensemble strategy is to use part of the training dataset to train an individual model each time and validate with the remaining part, while existing ensemble strategy uses all training data for each individual model. Our comprehensive empirical comparisons (Supplementary Tables 6 to 9) showed that the proposed ensemble strategy is better than existing ones.

Thank you very much for the response. I understand that the main aim of the work was to develop a deep learning model for a concrete medical problem. I was impressed by your multicenter study and the work of the experts who examined the data. Nevertheless, you used a simple deep learning method for classification, originating from a paper published several years ago. I don't see this as a very important issue (your method outperformed the experts after all), but given the latest developments in the field of deep learning, especially transfer learning, I believe it would be straightforward to improve your method. For example, you duplicated the US images into the RGB color channels to enable transfer learning, but it would be probably better to use a network trained on gray-scaled ImageNet images. You could also use image colorization, which would enable extraction of better performing features from the first blocks of the pre-trained model (there are no color gradients if you duplicate images). Moreover, it is unclear what transfer learning did you use (sorry if I missed this information). Did you fine-tune the entire network or perhaps several last convolutional blocks? Modern approaches to transfer learning include automatic selection of convolutional blocks to fine-tune.

Response: Yes, other transfer learning choices can be adopted here and may further improve the performance of the final ensemble model. While transfer learning is not the focus of the study and is largely independent of the proposed ensemble strategy, we performed several additional

experiments as suggested. In particular, gray-scaled ImageNet dataset was used to pretrain a SE-ResNet model, and then the model was finetuned for subsequent model training. The ensemble model based on such pre-trained SE-ResNet yields a sensitivity 85.6% and a specificity 90.7% (AUC 0.926) at image level, and a sensitivity 90.2% and a specificity 89.8% (AUC 0.934) at patient level. The diagnostic performance at both levels is slightly worse than that of the ensemble model (AUC 0.942 at image level, and 0.956 at patient level) reported in the manuscript which is based on the pretrained model from color ImageNet images. This is probably because color images contain more visual features than gray-scale images, and therefore the pretrained model from color images may have learned to represent more diversified visual features and is more generalizable to other tasks.

For the image colorization idea, we chose the colorization model pretrained on the 1.3M images from ImageNet with the colorization method (<http://richzhang.github.io/colorization/>), and applied it to the gray ultrasound images to generate colorized ultrasound images. Then the pretrained SE-ResNet model (based on color ImageNet images) was fine-tuned with the colorized ultrasound images for subsequent model training. In this way, the ensemble model yields a sensitivity 74.6% and a specificity 93.6% (AUC 0.876) at image level, and a sensitivity 79.4% and a specificity 92.3% (AUC 0.884) at patient level. These two AUCs are clearly lower than the result (AUC 0.942 at image level and 0.956 at patient level) obtained by the proposed ensemble strategy.

Regarding the fine-tuning detail for each individual model training, the entire network was fine-tuned with the same learning rate for all model layers. More advanced fine-tuning strategies, including the automatic selection of convolutional blocks to fine-tune, are left for further investigation considering the fine-tuning (or transfer learning) is not the focus of this study.

2) “In this way, we not only solved the issue about when to stop training a CNN, but also make the five trained CNNs a bit more diverse from each other, where the diversity among CNNs would improve the generalization ability of the ensembled model as confirmed in the empirical evaluation.” But this was not quantitatively examined. Perhaps you would obtain better results with an ensemble to 10 models? Moreover, each model forming the ensemble was based on the same network pre-trained on the ImageNet dataset. I would hardly expect any meaningful differences between the models forming the ensemble.”

Response: Thank you for the comments! We are sorry we did not show the empirical comparison in the first submission. Based on the comments, we added a series of empirical comparisons, including the results from (1) an individual CNN model trained with the entire training dataset, (2) the general ensemble model (of 5 and 10 individual models respectively) where each individual model is trained with the entire training dataset, and (3) different CNN backbones, all showing superior performance from the proposed ensemble strategy (Supplementary Tables 6 to 9). In our ensemble strategy, multiple individual models starting from the same pretrained network would result in different model parameters (therefore different individual models) because each individual model was trained with a different but partially overlapped subset of the training dataset. With such different individual models, the ensemble model would often perform better than single models. Actually, even using the same set of data to train multiple individual models, due to randomness of mini-batches during model training, all these individual models would result in different model parameters, which also would be able to form an ensemble model

better than individual ones. This has been confirmed in both previous studies with natural images and in our study here. Last but not least, the proposed ensemble strategy is expected to be directly applied to general intelligent diagnosis tasks, and we would release our source code once the study is ready to be published.

“Thank you for sharing Tables 6-9, but I am a little bit confused. Do you plan to include these tables in the final paper? If yes, please provide the descriptions of the experiments in the supplementary materials.”

Response: As suggested, we have added the descriptions of the additional experiments in the supplementary materials in the new version.

“If I understand correctly, you used the entire data for training instead of 5-fold cross-validation, which unsurprisingly resulted in worse performance.”

Response: In the previous (second) submission, we included a series of experiments for comparison. In particular, the entire training set was used for training of two different models. One is an individual CNN model (Model A), and the other is an ensemble model (of 5 and 10 individual models respectively) (Model B) where each individual model is trained with the entire training dataset. While it is expected Model A has relatively worse performance, it is not that obvious that Model B would also be expected worse than the proposed ensemble strategy, considering that both Model B and the proposed model are ensemble of multiple individual models.

“Moreover, other strategies to ensemble formation have not been investigated (as stated by the second reviewer). It would be interesting to check whether the ensemble of different deep learning models (listed in tables 6-9) leads to better classification.”

Response: We are sorry that we did not find more ensemble formations stated by the second reviewer this time. The suggested experiments from the second reviewer last time have been performed and included in the second submission. For the suggested ensemble of different deep learning models, we compared the diagnostic performance between the ensemble of five different models (SE_ResNet-152, DenseNet-201, EfficientNet-B7, ResNet-152 and VggNet-19) and the ensemble model of five SE_ResNet-152 models based on our proposed strategy. The ensemble of different deep learning models yields a sensitivity 80.5% and a specificity 93.9% (AUC 0.925) when trained by 5-fold cross validation, and yields a sensitivity 72.9 % and a specificity 97.0% (AUC 0.901) when trained with the entire dataset for each individual model. These two AUCs are clearly lower than the result (AUC 0.942) obtained by the proposed ensemble strategy.

Please, also clarify following issues:

- My initial concern has not been addressed. To form your ensemble, you applied the 5-fold validation, but perhaps 3-fold or 10-fold validation would work better?

Response: As suggested, we have added the 3-fold and 10-fold experiments, and the results are included in supplementary Tables 6-9. The diagnostic performance of the models trained with 3-fold and 10-fold validation is either lower or comparable to that of the model based on 5-fold validation. Besides, we would like to emphasize that this study proposes a better ensemble

framework (based on k-fold validation) than general ensemble strategies (based on entire dataset), and the number of folds in our ensemble strategy can be considered as a hyperparameter and experimentally determined for any specific application.

- “the training was stopped when the performance of the CNN started to decrease on the remaining subset.” but I could not find information how you stopped the training for the other methods (tables 6-9, training data for each individual model = All). Was there a stopping criterion, or perhaps you trained each model for 210 epochs, which the sup material suggests (‘The maximum number of epochs was set to 210 before which each model has been well trained without much change in model parameters’). If there was no stopping criterion, it is not surprising that the original ensemble method performed better. Stopping based on a separate dataset commonly result in better models.

Response: We are sorry that we did not clearly describe the stopping criteria for the other methods (Tables 6-9, training data for each individual model= All). Yes, each individual model was trained to the maximum number of epochs (i.e., 210 epochs). This is mainly because all training data has been used for the training of each individual model and there is no extra training data for additional validation. This is actually the weakness of using all training data for individual model training and also one motivation of applying the proposed ensemble strategy (e.g., with 4 out of 5 folds for each individual model).

3) “"Second, a pre-trained Se-ResNet based on the large-scale natural image dataset ImageNet was used to initialize the model parameters." ImageNet images are RGB, while US images are gray-scale. How did you adjust the pre-trained model to work with grayscale images?”

Response: Each single-channel grayscale image become a three-channel image by simply duplicating the original single channel three times. It is worth noting that this is a general routine for a pretrained AI model to process grayscale images.

Thank you. Could you add this information to the manuscript?

Response: as suggested, we have added the information in the supplementary materials.

11) “for an model ensemble, did you average activation maps obtained for each model to yield the final activation map?”

Response: No, an activation map was generated based on one single model rather than the ensembled model, because different single models in an ensemble model may have different prediction results, which would result in more or less different activation maps. The single model which has strong activation and the same prediction result as the ensembled model was chosen to generate the activation map, and the activation map is representative enough to show the attended region during ensemble model prediction.

Thank you. Please, include this information in the manuscript.

Response: A suggested, we have added the information in Figure 6.

Reviewer #2:

1) “This work is an excellent example of image processing tools applied to medical diagnosis. From a methodological point of view, however, the paper is of limited novelty and sophistication. In particular the option of using five CNN, trained by different data sets, with images randomly select from the main dataset, and averaging the results is questionable and need to be compared with other obvious strategies.”

Response: We agree that our study is application-oriented. However, we believe the proposed ensemble deep learning strategy is generalizable and better than state-of-the-art strategies. To support this claim, we have performed a series of experiments by considering (1) a single CNN model trained with the entire training dataset, (2) the general ensemble strategy where each single model is trained with the entire training dataset, and (3) different CNN backbones, all showing superior performance from the proposed ensemble strategy (Supplementary material Table 6 to 9). What’s more, the implementation is not sophisticated, and we plan to release our source code once the study is ready to be published. The proposed ensemble strategy is expected to be directly applied to general intelligent diagnosis tasks.

Thank you for the response. Please, see the first and the second response of the reviewer #1.

Response: We have added experiments as suggested in the first and second response for Reviewer 1.

3) “The idea of conjugating the automatic classification with the classification of the expert is very interesting but in practice this corresponds to the normal strategy of any classical CAD (Computer aided Diagnosis) system. What the authors prove is that the proposed system is able to help the expert to improve the diagnosis.”

Response: Yes! One purpose of this study is to build a deep learning model to help experts/doctors improve their diagnosis of biliary atresia.

Thank you. Notice, however, that it would be interesting to first select the US images misdiagnosed by the experts, and train the machine learning model to pay more attention to correct classification of these cases.

Response: This is an interesting idea! However, (1) if the images misdiagnosed by the experts were from the training set, the machine learning can already correctly learn to recognize them because the model is trained to correctly recognize all training images; (2) if the misdiagnosis happens during testing rather than training, it is difficult for the machine learning model to know in advance which test images would be misdiagnosed by experts. Also, all test images should not be used to fine-tune the machine learning models and therefore the misdiagnosed (test) images by experts would not affect the training of the machine learning model. One possible future study may explore the continual update of the machine learning model based on the misdiagnosed images by specific experts, which is beyond the scope of this study.

5) “The ROC curves of the model at both levels also confirmed its superior performance over human experts (Figure 1)” . How are the ROC curves displayed in Fig.1, 2 and 3 and referred in line 135 computed?

Response: Basically, each ROC was generated by varying the threshold for the output of the model. More details were in the “Materials and Methods: Measurements of the diagnostic

performance” section.

Thank you. But did you use trapezoidal approach for calculations? How did you compute AUCs for the experts (single point)?

Response: Yes. For the experts who only have a single point, trapezoidal approach was used for calculations of AUCs. Specifically, we connected the single point of the expert with a point (0, 0) and a point (1, 1) to form a broken line, and then calculated the area under the broken line as follow: $AUC = (\text{sensitivity} + \text{specificity})/2$. The calculations were performed with MedCalc Statistical Software version 15.2.2 (MedCalc Software bvba, Ostend, Belgium).

6) “Line: 162: “Furthermore, when using images of mediocre quality (with frequency < 14MHz, scanning period ≤ 2018 , supersonic + others or Mindary + others) ” . However in line 420 it is suggest you are using “high-frequency transducers (> 7 MHz) ” which may suggest you may are using images of mediocre quality. Is that right?”

Response: Not really. In general, transducers with higher frequency would produce images in higher quality. However, the quality of images can be influenced by other parameters as well. Therefore, the quality of images was not merely determined by the frequency of transducers. Actually, all images included in the study were qualified for further analysis after an expert screened all images and excluded unqualified images. The aim of dividing training data into different groups for further analysis was to evaluate generalization of the model. Although the quality of images with frequency $\geq 14\text{MHz}$ is better than that with frequency $< 14\text{MHz}$, it did not mean the images with frequency $< 14\text{MHz}$ is mediocre. We noticed that the word ‘mediocre’ could be inappropriate and misleading and therefore has been discarded in the new manuscript (instead, using words like ‘images in relatively moderate quality’).

Thank you. Could you also include the information about the number of cases for each subgroup in Sup Table 2?

Response: As suggested, we have added the number of images and patients for each subgroup in Sup Table 1 and Table 2 respectively.

Additional comments:

- Please, proofread the manuscript. It includes some grammatical errors and awkward sentences.

Response: We have proofread the manuscript and tried to correct grammatical errors and rephrase certain sentences.

- Fig. 8a) suggests that you averaged the training set predictions to obtain diagnostic results presented in Table 1. The performance on the training set is commonly too optimistic and should not be reported. Please, clarify.

Response: Table 1 is about the internal evaluation result. The reported performance of the deep learning model is not on the training set, instead on five cross-validation subsets, each used as a ‘test set’ which was not used to train the corresponding ensemble model. Specifically, every time four of the five subsets were used as a training dataset to train an ensemble deep learning model (i.e., dividing the four subsets into 5 new subsets and then following Figure 8a for ensemble

model training), and the ensembled model was then applied to predict the category of each image in the remaining one (test) subset. Such a process was repeated five times, each time using a unique subset as the test dataset. Therefore, the reported deep learning performance in Table 1 should not be considered ‘too optimistic’.

- If I understand correctly, you did not apply the cross-validation to obtain the results presented in supp Table 1 and 2. What was the training procedure in this case?

Response: We actually applied the 5-fold cross-validation procedure, as explained in the response to the previous comment. The training method was shown in Figure 8 and described in the corresponding main text.

- ‘The rate of dropout was set to 0.2 for all tests in this study’ . Do you mean for the training of the network? Please, specify whether you applied the dropout to all layers of the network.

Response: Yes, dropout was used for the training of network models. Dropout was applied to only the last fully connected layer, which has been described in the manuscript.

- Please, better describe the transfer learning method. Clearly indicate which convolutional blocks were used for fine-tuning etc

Response: Please refer to the response to first comment from Reviewer 1. Basically, all layers were fine-tuned.

- “The mean and standard deviation of pixel intensities were computed over all cropped images, which were then used to normalize each pixel in all rescaled images. Each normalized image will be used as an input to the SE-ResNet for model training.” But the SE-ResNet paper states ‘Each input image is normalised through mean RGB-channel subtraction.’ Why did you apply different normalization?

Response: We adopted the normalization based on both the mean and the standard deviation because it has been widely used as a preprocessing step for training of most types of deep convolutional models. More important, the pretrained SE-ResNet we employed also applies the normalization based on both the mean and standard deviation, therefore we have to use this normalization method to be consistent with the normalization choice for the original training of the pretrained model.

- Fig. 8a), there is an issue with the prediction and set numeration.

Response: We have modified the numeration issue in Figure 8a as suggested.

- Fig. 8b), I would suggest to improve the resolution of the images. It is impossible to read the text, even after zooming in.

Response: We have replaced it by a high-resolution image as suggested.

- From the sup material: ‘To improve the generalizability of each SE-ResNet, besides the ensembled learning mentioned above’, perhaps it would be better to replace ‘above’ with ‘in the main text’ ?

Response: We have replaced it as suggested.

Reviewer #3:

The revised version is much improved. As for me, it would have been useful to compare the results with a group of infants with intrahepatic cholestasis.

Response: Thank you for the suggestion! The main objective of this study is to train an ensemble deep learning model to distinguish infants with BA from infants with jaundice, and the majority of infants with jaundice in this study were actually infants with intrahepatic cholestasis. Although a small number of healthy infants were also included to expand the sample size, they were the minority in this study and thus the infants with jaundice but not BA and the (small number of) healthy infants were grouped as the infants without BA in the study. In addition, a group of infants with intrahepatic cholestasis were included when testing in the smartphone app. The results showed that the ensemble deep learning model could identify infants with BA from infants with intrahepatic cholestasis, with accuracy 85.6%.

Reviewer #4 comments

The responses to all the questions in the previous review have been answered satisfactorily, except q 4. It is not clear in the revised manuscript what these experiments were.

q2) why was expert voting, even for the ratings of 2 experts on the validation set, not done? Ideally, the same strategy as described in the answer to q9 should have been used to pool the diagnoses of both the experts.

Overall, I got the impression that this manuscript needs to change its stated claim of outperforming human experts to aiding diagnosis accuracy.

Dear Reviewers,

Thank you for the very valuable comments and suggestions again! As suggested, we have performed additional experiments and the results were presented in the manuscript and partly in the response below. We also revised the manuscript (in red) accordingly.

The detailed response to each comment is listed below. Only the comments requiring further revision were included, for which all the corresponding previous responses and the previous comments were also kept for better understanding of the latest comments and the responses (in red).

Reviewer #1:

1) “there is little to no novelty in respect to machine learning methods. The authors used well known network architecture and simple transfer learning method to develop the models. The same stands for the methods used to combine predictions of the experts and the models.”

Response: We would like to emphasize that the main objective of this study is not to develop a novel deep learning approach but to apply deep learning technique to an important medical problem. The expert-level AI model developed in this study and the combined human-model diagnosis would help radiologists largely improve their diagnosis performance particularly in undeveloped regions. On the other hand, although the proposed ensemble model is based on the existing ensemble strategy, the (slight) novelty of the proposed ensemble strategy is to use part of the training dataset to train an individual model each time and validate with the remaining part, while existing ensemble strategy uses all training data for each individual model. Our comprehensive empirical comparisons (Supplementary Tables 6 to 9) showed that the proposed ensemble strategy is better than existing ones.

Thank you very much for the response. I understand that the main aim of the work was to develop a deep learning model for a concrete medical problem. I was impressed by your multicenter study and the work of the experts who examined the data. Nevertheless, you used a simple deep learning method for classification, originating from a paper published several years ago. I don't see this as a very important issue (your method outperformed the experts after all), but given the latest developments in the field of deep learning, especially transfer learning, I believe it would be straightforward to improve your method. For example, you duplicated the US images into the RGB color channels to enable transfer learning, but it would be probably better to use a network trained on gray-scaled ImageNet images. You could also use image colorization, which would enable extraction of better performing features from the first blocks of the pre-trained model (there are no color gradients if you duplicate images). Moreover, it is unclear what transfer learning did you use (sorry if I missed this information). Did you fine-tune the entire network or perhaps several last convolutional blocks? Modern approaches to transfer learning include automatic selection of convolutional blocks to fine-tune.

Response: Yes, other transfer learning choices can be adopted here and may further improve the performance of the final ensemble model. While transfer learning is not the focus of the study and is largely independent of the proposed ensemble strategy, we performed several additional

experiments as suggested. In particular, gray-scaled ImageNet dataset was used to pretrain a SE-ResNet model, and then the model was finetuned for subsequent model training. The ensemble model based on such pre-trained SE-ResNet yields a sensitivity 85.6% and a specificity 90.7% (AUC 0.926) at image level, and a sensitivity 90.2% and a specificity 89.8% (AUC 0.934) at patient level. The diagnostic performance at both levels is slightly worse than that of the ensemble model (AUC 0.942 at image level, and 0.956 at patient level) reported in the manuscript which is based on the pretrained model from color ImageNet images. This is probably because color images contain more visual features than gray-scale images, and therefore the pretrained model from color images may have learned to represent more diversified visual features and is more generalizable to other tasks.

For the image colorization idea, we chose the colorization model pretrained on the 1.3M images from ImageNet with the colorization method (<http://richzhang.github.io/colorization/>), and applied it to the gray ultrasound images to generate colorized ultrasound images. Then the pretrained SE-ResNet model (based on color ImageNet images) was fine-tuned with the colorized ultrasound images for subsequent model training. In this way, the ensemble model yields a sensitivity 74.6% and a specificity 93.6% (AUC 0.876) at image level, and a sensitivity 79.4% and a specificity 92.3% (AUC 0.884) at patient level. These two AUCs are clearly lower than the result (AUC 0.942 at image level and 0.956 at patient level) obtained by the proposed ensemble strategy.

Regarding the fine-tuning detail for each individual model training, the entire network was fine-tuned with the same learning rate for all model layers. More advanced fine-tuning strategies, including the automatic selection of convolutional blocks to fine-tune, are left for further investigation considering the fine-tuning (or transfer learning) is not the focus of this study.

2) “In this way, we not only solved the issue about when to stop training a CNN, but also make the five trained CNNs a bit more diverse from each other, where the diversity among CNNs would improve the generalization ability of the ensembled model as confirmed in the empirical evaluation.” But this was not quantitatively examined. Perhaps you would obtain better results with an ensemble to 10 models? Moreover, each model forming the ensemble was based on the same network pre-trained on the ImageNet dataset. I would hardly expect any meaningful differences between the models forming the ensemble.”

Response: Thank you for the comments! We are sorry we did not show the empirical comparison in the first submission. Based on the comments, we added a series of empirical comparisons, including the results from (1) an individual CNN model trained with the entire training dataset, (2) the general ensemble model (of 5 and 10 individual models respectively) where each individual model is trained with the entire training dataset, and (3) different CNN backbones, all showing superior performance from the proposed ensemble strategy (Supplementary Tables 6 to 9). In our ensemble strategy, multiple individual models starting from the same pretrained network would result in different model parameters (therefore different individual models) because each individual model was trained with a different but partially overlapped subset of the training dataset. With such different individual models, the ensemble model would often perform better than single models. Actually, even using the same set of data to train multiple individual models, due to randomness of mini-batches during model training, all these individual models would result in different model parameters, which also would be able to form an ensemble model

better than individual ones. This has been confirmed in both previous studies with natural images and in our study here. Last but not least, the proposed ensemble strategy is expected to be directly applied to general intelligent diagnosis tasks, and we would release our source code once the study is ready to be published.

“Thank you for sharing Tables 6-9, but I am a little bit confused. Do you plan to include these tables in the final paper? If yes, please provide the descriptions of the experiments in the supplementary materials.”

Response: As suggested, we have added the descriptions of the additional experiments in the supplementary materials in the new version.

“If I understand correctly, you used the entire data for training instead of 5-fold cross-validation, which unsurprisingly resulted in worse performance.”

Response: In the previous (second) submission, we included a series of experiments for comparison. In particular, the entire training set was used for training of two different models. One is an individual CNN model (Model A), and the other is an ensemble model (of 5 and 10 individual models respectively) (Model B) where each individual model is trained with the entire training dataset. While it is expected Model A has relatively worse performance, it is not that obvious that Model B would also be expected worse than the proposed ensemble strategy, considering that both Model B and the proposed model are ensemble of multiple individual models.

“Moreover, other strategies to ensemble formation have not been investigated (as stated by the second reviewer). It would be interesting to check whether the ensemble of different deep learning models (listed in tables 6-9) leads to better classification.”

Response: We are sorry that we did not find more ensemble formations stated by the second reviewer this time. The suggested experiments from the second reviewer last time have been performed and included in the second submission. For the suggested ensemble of different deep learning models, we compared the diagnostic performance between the ensemble of five different models (SE_ResNet-152, DenseNet-201, EfficientNet-B7, ResNet-152 and VggNet-19) and the ensemble model of five SE_ResNet-152 models based on our proposed strategy. The ensemble of different deep learning models yields a sensitivity 80.5% and a specificity 93.9% (AUC 0.925) when trained by 5-fold cross validation, and yields a sensitivity 72.9 % and a specificity 97.0% (AUC 0.901) when trained with the entire dataset for each individual model. These two AUCs are clearly lower than the result (AUC 0.942) obtained by the proposed ensemble strategy.

Please, also clarify following issues:

- My initial concern has not been addressed. To form your ensemble, you applied the 5-fold validation, but perhaps 3-fold or 10-fold validation would work better?

Response: As suggested, we have added the 3-fold and 10-fold experiments, and the results are included in supplementary Tables 6-9. The diagnostic performance of the models trained with 3-fold and 10-fold validation is either lower or comparable to that of the model based on 5-fold validation. Besides, we would like to emphasize that this study proposes a better ensemble

framework (based on k-fold validation) than general ensemble strategies (based on entire dataset), and the number of folds in our ensemble strategy can be considered as a hyperparameter and experimentally determined for any specific application.

- “the training was stopped when the performance of the CNN started to decrease on the remaining subset.” but I could not find information how you stopped the training for the other methods (tables 6-9, training data for each individual model = All). Was there a stopping criterion, or perhaps you trained each model for 210 epochs, which the sup material suggests (‘The maximum number of epochs was set to 210 before which each model has been well trained without much change in model parameters’). If there was no stopping criterion, it is not surprising that the original ensemble method performed better. Stopping based on a separate dataset commonly result in better models.

Response: We are sorry that we did not clearly describe the stopping criteria for the other methods (Tables 6-9, training data for each individual model= All). Yes, each individual model was trained to the maximum number of epochs (i.e., 210 epochs). This is mainly because all training data has been used for the training of each individual model and there is no extra training data for additional validation. This is actually the weakness of using all training data for individual model training and also one motivation of applying the proposed ensemble strategy (e.g., with 4 out of 5 folds for each individual model).

3) “"Second, a pre-trained Se-ResNet based on the large-scale natural image dataset ImageNet was used to initialize the model parameters." ImageNet images are RGB, while US images are gray-scale. How did you adjust the pre-trained model to work with grayscale images?”

Response: Each single-channel grayscale image become a three-channel image by simply duplicating the original single channel three times. It is worth noting that this is a general routine for a pretrained AI model to process grayscale images.

Thank you. Could you add this information to the manuscript?

Response: as suggested, we have added the information in the supplementary materials.

11) “for an model ensemble, did you average activation maps obtained for each model to yield the final activation map?”

Response: No, an activation map was generated based on one single model rather than the ensemble model, because different single models in an ensemble model may have different prediction results, which would result in more or less different activation maps. The single model which has strong activation and the same prediction result as the ensemble model was chosen to generate the activation map, and the activation map is representative enough to show the attended region during ensemble model prediction.

Thank you. Please, include this information in the manuscript.

Response: A suggested, we have added the information in Figure 6.

Reviewer #2:

1) “This work is an excellent example of image processing tools applied to medical diagnosis. From a methodological point of view, however, the paper is of limited novelty and sophistication. In particular the option of using five CNN, trained by different data sets, with images randomly select from the main dataset, and averaging the results is questionable and need to be compared with other obvious strategies.”

Response: We agree that our study is application-oriented. However, we believe the proposed ensemble deep learning strategy is generalizable and better than state-of-the-art strategies. To support this claim, we have performed a series of experiments by considering (1) a single CNN model trained with the entire training dataset, (2) the general ensemble strategy where each single model is trained with the entire training dataset, and (3) different CNN backbones, all showing superior performance from the proposed ensemble strategy (Supplementary material Table 6 to 9). What’s more, the implementation is not sophisticated, and we plan to release our source code once the study is ready to be published. The proposed ensemble strategy is expected to be directly applied to general intelligent diagnosis tasks.

Thank you for the response. Please, see the first and the second response of the reviewer #1.

Response: We have added experiments as suggested in the first and second response for Reviewer 1.

3) “The idea of conjugating the automatic classification with the classification of the expert is very interesting but in practice this corresponds to the normal strategy of any classical CAD (Computer aided Diagnosis) system. What the authors prove is that the proposed system is able to help the expert to improve the diagnosis.”

Response: Yes! One purpose of this study is to build a deep learning model to help experts/doctors improve their diagnosis of biliary atresia.

Thank you. Notice, however, that it would be interesting to first select the US images misdiagnosed by the experts, and train the machine learning model to pay more attention to correct classification of these cases.

Response: This is an interesting idea! However, (1) if the images misdiagnosed by the experts were from the training set, the machine learning can already correctly learn to recognize them because the model is trained to correctly recognize all training images; (2) if the misdiagnosis happens during testing rather than training, it is difficult for the machine learning model to know in advance which test images would be misdiagnosed by experts. Also, all test images should not be used to fine-tune the machine learning models and therefore the misdiagnosed (test) images by experts would not affect the training of the machine learning model. One possible future study may explore the continual update of the machine learning model based on the misdiagnosed images by specific experts, which is beyond the scope of this study.

5) “The ROC curves of the model at both levels also confirmed its superior performance over human experts (Figure 1)” . How are the ROC curves displayed in Fig.1, 2 and 3 and referred in line 135 computed?

Response: Basically, each ROC was generated by varying the threshold for the output of the model. More details were in the “Materials and Methods: Measurements of the diagnostic

performance” section.

Thank you. But did you use trapezoidal approach for calculations? How did you compute AUCs for the experts (single point)?

Response: Yes. For the experts who only have a single point, trapezoidal approach was used for calculations of AUCs. Specifically, we connected the single point of the expert with a point (0, 0) and a point (1, 1) to form a broken line, and then calculated the area under the broken line as follow: $AUC = (\text{sensitivity} + \text{specificity})/2$. The calculations were performed with MedCalc Statistical Software version 15.2.2 (MedCalc Software bvba, Ostend, Belgium).

6) “Line: 162: “Furthermore, when using images of mediocre quality (with frequency < 14MHz, scanning period ≤ 2018 , supersonic + others or Mindary + others) ” . However in line 420 it is suggest you are using “high-frequency transducers (> 7 MHz) ” which may suggest you may are using images of mediocre quality. Is that right?”

Response: Not really. In general, transducers with higher frequency would produce images in higher quality. However, the quality of images can be influenced by other parameters as well. Therefore, the quality of images was not merely determined by the frequency of transducers. Actually, all images included in the study were qualified for further analysis after an expert screened all images and excluded unqualified images. The aim of dividing training data into different groups for further analysis was to evaluate generalization of the model. Although the quality of images with frequency $\geq 14\text{MHz}$ is better than that with frequency $< 14\text{MHz}$, it did not mean the images with frequency $< 14\text{MHz}$ is mediocre. We noticed that the word ‘mediocre’ could be inappropriate and misleading and therefore has been discarded in the new manuscript (instead, using words like ‘images in relatively moderate quality’).

Thank you. Could you also include the information about the number of cases for each subgroup in Sup Table 2?

Response: As suggested, we have added the number of images and patients for each subgroup in Sup Table 1 and Table 2 respectively.

Additional comments:

- Please, proofread the manuscript. It includes some grammatical errors and awkward sentences.

Response: We have proofread the manuscript and tried to correct grammatical errors and rephrase certain sentences.

- Fig. 8a) suggests that you averaged the training set predictions to obtain diagnostic results presented in Table 1. The performance on the training set is commonly too optimistic and should not be reported. Please, clarify.

Response: Table 1 is about the internal evaluation result. The reported performance of the deep learning model is not on the training set, instead on five cross-validation subsets, each used as a ‘test set’ which was not used to train the corresponding ensemble model. Specifically, every time four of the five subsets were used as a training dataset to train an ensemble deep learning model (i.e., dividing the four subsets into 5 new subsets and then following Figure 8a for ensemble

model training), and the ensembled model was then applied to predict the category of each image in the remaining one (test) subset. Such a process was repeated five times, each time using a unique subset as the test dataset. Therefore, the reported deep learning performance in Table 1 should not be considered ‘too optimistic’.

- If I understand correctly, you did not apply the cross-validation to obtain the results presented in supp Table 1 and 2. What was the training procedure in this case?

Response: We actually applied the 5-fold cross-validation procedure, as explained in the response to the previous comment. The training method was shown in Figure 8 and described in the corresponding main text.

- ‘The rate of dropout was set to 0.2 for all tests in this study’ . Do you mean for the training of the network? Please, specify whether you applied the dropout to all layers of the network.

Response: Yes, dropout was used for the training of network models. Dropout was applied to only the last fully connected layer, which has been described in the manuscript.

- Please, better describe the transfer learning method. Clearly indicate which convolutional blocks were used for fine-tuning etc

Response: Please refer to the response to first comment from Reviewer 1. Basically, all layers were fine-tuned.

- “The mean and standard deviation of pixel intensities were computed over all cropped images, which were then used to normalize each pixel in all rescaled images. Each normalized image will be used as an input to the SE-ResNet for model training.” But the SE-ResNet paper states ‘Each input image is normalised through mean RGB-channel subtraction.’ Why did you apply different normalization?

Response: We adopted the normalization based on both the mean and the standard deviation because it has been widely used as a preprocessing step for training of most types of deep convolutional models. More important, the pretrained SE-ResNet we employed also applies the normalization based on both the mean and standard deviation, therefore we have to use this normalization method to be consistent with the normalization choice for the original training of the pretrained model.

- Fig. 8a), there is an issue with the prediction and set numeration.

Response: We have modified the numeration issue in Figure 8a as suggested.

- Fig. 8b), I would suggest to improve the resolution of the images. It is impossible to read the text, even after zooming in.

Response: We have replaced it by a high-resolution image as suggested.

- From the sup material: ‘To improve the generalizability of each SE-ResNet, besides the ensembled learning mentioned above’, perhaps it would be better to replace ‘above’ with ‘in the main text’ ?

Response: We have replaced it as suggested.

Reviewer #3:

The revised version is much improved. As for me, it would have been useful to compare the results with a group of infants with intrahepatic cholestasis.

Response: Thank you for the suggestion! The main objective of this study is to train an ensemble deep learning model to distinguish infants with BA from infants with jaundice, and the majority of infants with jaundice in this study were actually infants with intrahepatic cholestasis. Although a small number of healthy infants were also included to expand the sample size, they were the minority in this study and thus the infants with jaundice but not BA and the (small number of) healthy infants were grouped as the infants without BA in the study. In addition, a group of infants with intrahepatic cholestasis were included when testing in the smartphone app. The results showed that the ensemble deep learning model could identify infants with BA from infants with intrahepatic cholestasis, with accuracy 85.6%.

Reviewer #4 comments

q1) The responses to all the questions in the previous review have been answered satisfactorily, except q 4. It is not clear in the revised manuscript what these experiments were.

Response: Thanks for your comments. These experiments were summarized in Supplementary Tables 6-9 and corresponding descriptions in the Supplementary Materials.

q2) why was expert voting, even for the ratings of 2 experts on the validation set, not done? Ideally, the same strategy as described in the answer to q9 should have been used to pool the diagnoses of both the experts.

Responses: In the study, we compared the combination of the AI model and an individual expert with the individual expert in BA diagnosis. The purpose is to evaluate whether the addition of the AI model would help improve the diagnosis performance of any expert. While it is interesting to compare the AI+expert with expert+expert, it is not practical to ask two experts to diagnose together in real applications, particularly considering that BA is a rare disease and the number of BA experts is very limited.

q3) Overall, I got the impression that this manuscript needs to change its stated claim of outperforming human experts to aiding diagnosis accuracy.

Response: We agree with you. In fact, in the **Abstract** and **Conclusion**, we claimed that the ensembled deep learning model provides a solution to help radiologists improve BA diagnosis, which is similar to aiding diagnosis accuracy.

We sincerely hope our reponse and revision will meet you the quality of publishment in your journal of NATURE COMMUNICATIONS.

Best Regards,

Luyao Zhou, MD,Ph.D

Reviewers' Comments:

Reviewer #1:

Remarks to the Author:

Manuscript looks much better now. The authors have improved the manuscript and accordingly addressed the main issues I had raised in the previous review. There are, however, several minor issues that still remain. Please, find my specific comments below:

- I suggest to proofread the manuscript. It still includes some grammatical errors.
- "Table 1. The diagnostic performance of the ensembled deep learning model and two human experts on the internal cross-validation dataset." If I understand correctly, the performance of the experts was not calculated by means of cross-validation. Please, clarify table description.
- Please, include information about the method used to calculate 95% confidence intervals.
- "The single model which was within the EDLM and had strong activation was chosen to generate the activation map." Could you please be more precise?
- I still think that Fig. 8b is a little bit confusing. If I understand correctly, the averaging procedure is related to the images from the external dataset. To calculate the performance on the first dataset (n=3705), you did not apply averaging. Unfortunately, to some readers Fig. 8b may suggest that you averaged results obtained using training images from the first dataset, somehow violating the cross-validation. Perhaps it would be good to modify the figure.
- "time point 515 to stop the training of each CNN was unique (e.g., subset 5 for first CNN, and subset 1 for 516 second CNN)" Fig. 8b suggests that subset 1 was used for the first CNN.

Supp materials:

- "The mean and standard deviation of pixel intensities were computed over all cropped images, which were then used to normalize each pixel in all rescaled images. The normalization method was consistent with the pretrained model used." Shouldn't you provide a reference? Seems that this normalization is different than the one originally used by the authors of the SE-ResNet paper. Could you please clarify?

"a pre-trained SE-ResNet based on the large-scale natural image dataset ImageNet was used to initialize the model parameters" Did you use a pre-trained network from a public repository? If yes, then provide a reference. Or perhaps did you download the imagenet dataset and pre-train the network yourself (using a different normalization)? Please, clarify in the text and double check the normalization method. Moreover, if I understand correctly, the last dense layer was not initialized with the ImageNet weights, but trained from scratch to do binary classification. Please, also include this information.

- Please, indicate what software and deep learning libraries were used to develop the methods.

Reviewer #4:

Remarks to the Author:

The revised version is generally fine, and addresses my earlier comments and so I can now recommend acceptance.

Thank you again for your positive comments on this manuscript. As suggested, we have revised the manuscript (in red) accordingly. The detailed response to each comment is listed below (in red).

Reviewer #1:

- I suggest to proofread the manuscript. It still includes some grammatical errors.

Response: Thank you for the suggestion! We have proofread this manuscript again and tried our best to correct grammatical errors and rephrased certain sentences.

- “Table 1. The diagnostic performance of the ensembled deep learning model and two human experts on the internal cross-validation dataset.” If I understand correctly, the performance of the experts was not calculated by means of cross-validation. Please, clarify table description.

Response: Thank you for the correction! We have clarified the table description.

- Please, include information about the method used to calculate 95% confidence intervals.

Response: As suggested, we have included relevant information in the “Materials and Methods: Measurements of the diagnostic performance” section.

- “The single model which was within the EDLM and had strong activation was chosen to generate the activation map.” Could you please be more precise?

Response: There were five activation maps generated by 5 individual models within the EDLM for each image in the external validation dataset. Within the activation maps whose associated individual models have the same classification result as that of the ensemble deep learning model, the activation map which had the highest mean activation was selected for consistency assessment in comparison with human experts. We have included the description in the revised manuscript.

- I still think that Fig. 8b is a little bit confusing. If I understand correctly, the averaging procedure is related to the images from the external dataset. To calculate the performance on the first dataset (n=3705), you did not apply averaging. Unfortunately, to some readers Fig. 8b may suggest that you averaged results obtained using training images from the first dataset, somehow violating the cross-validation. Perhaps it would be good to modify the figure.

Response: You might be referring to Figure 8a instead of Figure 8b? Actually, the figure can be used for both internal and external validation. Although the training cohort is the whole internal dataset (3705 images) for the external validation, the training cohort for each of the five internal cross-validation datasets is the remaining four subsets. That means, for the internal validation, every time four of the subsets were used as a training cohort (which is divided into five new subsets) to train an ensembled deep learning model, and the ensembled model was then applied to predict the category of each image in the remaining one (internal validation) subset. Such a process was repeated five times, each time using a unique subset as the internal validation dataset. We have included the above description and slightly modified Figure 8a for clarification.

- “time point 515 to stop the training of each CNN was unique (e.g., subset 5 for first CNN, and subset 1 for 516 second CNN)” Fig. 8b suggests that subset 1 was used for the first CNN.

Response: Thank you for the correction! We have corrected the number issue in the new manuscript.

Supp materials:

- “The mean and standard deviation of pixel intensities were computed over all cropped images, which were then used to normalize each pixel in all rescaled images. The normalization method was consistent with the pretrained model used.” Shouldn’t you provide a reference? Seems that this normalization is different than the one originally used by the authors of the SE-ResNet paper. Could you please clarify?

Response: As you mentioned, the normalization of our method is different from the one in original SE-ResNet paper, because the pre-trained model we used comes from one of the most commonly used repositories (<https://github.com/Cadene/pretrained-models.pytorch>) on github which have 7.6K stars. According to line 75 in the code released by the author (<https://github.com/Cadene/pretrained-models.pytorch/blob/master/pretrainedmodels/utils.py>), we found the input normalization for the pre-trained model is based on the mean and standard deviation of pixel intensities as described. To be consistent, we used the same normalization method which is also widely used in training other deep learning models. The github link and relevant description have been included in the supplementary material.

“a pre-trained SE-ResNet based on the large-scale natural image dataset ImageNet was used to initialize the model parameters” Did you use a pre-trained network from a public repository? If yes, then provide a reference. Or perhaps did you download the imagenet dataset and pre-train the network yourself (using a different normalization)? Please, clarify in the text and double check the normalization method. Moreover, if I understand correctly, the last dense layer was not initialized with the ImageNet weights, but trained from scratch to do binary classification. Please, also include this information.

Response: We used a pre-trained network from a github repository (<https://github.com/Cadene/pretrained-models.pytorch>). Yes, the last fully connection layer was initialized randomly, which is commonly used when starting from a pre-trained model. We have included the details in the supplementary material.

- Please, indicate what software and deep learning libraries were used to develop the methods.

Response: The libraries we used to develop the method have been included in the supplementary material.

Reviewer #4:

The revised version is generally fine, and addresses my earlier comments and so I can now

recommend acceptance.

Response: Thank you for your previous comments and suggestions!

We sincerely hope our reponse and revision will meet you the quality of publishment in your journal of NATURE COMMUNICATIONS.

Best Regards,

Luyao Zhou, MD,Ph.D

Reviewers' Comments:

Reviewer #1:

Remarks to the Author:

Authors have addressed all the issues I raised in the review. I have no more comments.

Thank you again for your positive comments on this manuscript. As suggested, we have revised the manuscript (in red) accordingly. The detailed response to each comment is listed below (in red).

REVIEWERS' COMMENTS

Reviewer #1 (Remarks to the Author):

Authors have addressed all the issues I raised in the review. I have no more comments.

Response: Thank you for all your comments and suggestions!

We sincerely hope our reponse and revision will meet you the quality of publishment in your journal of NATURE COMMUNICATIONS.

Best Regards,

Luyao Zhou, MD,Ph.D